# ON THE IMPACT OF MACHINE LEARNING RANDOMNESS ON GROUP FAIRNESS

## ABSTRACT

Statistical measures for group fairness in machine learning reflect the gap in performance of algorithms across different groups. These measures, however, exhibit a high variance, between different training instances, that makes them unreliable for empirical evaluation of fairness. What is the cause of this variance, and how can we reduce it? We investigate the impact of different sources of randomness in machine learning on group fairness. We show that the variance in group fairness measures is mainly due to the high volatility of the learning process on under-represented groups, which itself is largely caused by the stochasticity of data order during training. Based on these findings, we show how to manipulate group level accuracy (i.e. model fairness), with high efficiency and negligible impact on the overall predictive power of the model, by changing the data order.

## 1 INTRODUCTION

Machine learning models are shown to manifest and escalate historical prejudices and biases present in their training data (Crawford, 2013; Barocas & Selbst, 2016; Zhao et al., 2017; Abbasi et al., 2019). Understanding these biases and the following ethical considerations has led to the rise of fair machine learning research (Chouldechova & Roth, 2018; Caton & Haas, 2020; Mehrabi et al., 2021). Recent work in fair deep learning have observed a trend of high variance in fairness measures across multiple training runs (Qian et al., 2021; Amir et al., 2021; Sellam et al., 2021; Soares et al., 2022), usually attributed to non-determinism in training. These results have challenged the legitimacy of existing claims in the literature (Soares et al., 2022), and have even disputed the effectiveness of various bias mitigation techniques (Amir et al., 2021; Sellam et al., 2021). Thus, a reliable extraction of fairness trends in a model requires accounting for the high variance to avoid lottery winners (see Figure 1).

The naive solution of executing a large number of identical training runs to capture the overall variance creates a huge computational demand, and discourages the examination of biases in several rapidly growing forefronts of machine learning research by increasing the resource requirements. Therefore, understanding the actual cause of the high variance in the fairness measures is critical. To the best of our knowledge, we are the first to study fairness variance beyond trivially executing multiple identical training runs. More specifically, we show the following:

- We show that the trends of fairness variance observed in literature are dominated by random data reshuffling during training, which causes high fairness variance between epochs even within a single training run, while the non-determinism in weight initialization has minimal influence.

- We extract an empirical relationship between group representation and instability in group level performance, highlighting higher vulnerability of minority to changing model behavior. Our results attribute the high fairness variance to lower prediction stability for under-represented subgroups.

- We demonstrate an immediate dominance of the data order on model fairness. A model's fairness is predictable, based on only the most recent training points, irrespective of preceding model behavior.

- Based on this information, we propose to use the fairness variance across epochs as a proxy to study the changing model fairness across multiple training runs, thus reducing the computational requirements by a significant margin.

- Finally, we manipulate group level performances (i.e., model fairness) by changing the data order, with a relatively minor impact on the overall accuracy. This manipulation can improve fairness as well as reverse the effects of several bias mitigation algorithms within a single training epoch.

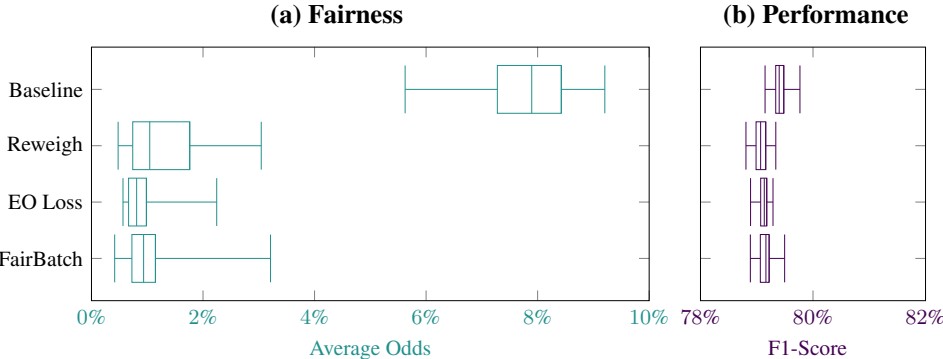

Figure 1: **(a)** Fairness (average odds) has a high variance across identical runs due to non-determinism in training. This variance persists even with several state-of-the-art bias mitigation algorithms (*Reweighing (Kamiran & Calders, 2012); Equalized Odds Loss (Fukuchi et al., 2020); FairBatch (Roh et al., 2020)*), and reliable comparison between these methods can only be made after capturing the overall model behavior, due to intersecting ranges. **(b)** The performance score (F1 score), however, has significantly smaller range of variance and doesn't face similar issues as fairness.

## 2 EXPERIMENTAL SETUP

**Datasets and Models**  We will conduct our investigation on ACSIncome and ACSEmployment tasks of the Folktables dataset (Ding et al., 2021), and binary classification of the 'smiling' label in CelebA dataset (Liu et al., 2015), with gender (Female vs. Male) as the sensitive attribute for all datasets. For CelebA, input features are obtained by passing the input image through a pre-trained and frozen ResNet-50 backbone and extracting the output feature vector after the final average pooling layer. More details on the datasets are provided in Appendix A.

We train a feed forward network on input features with a single hidden layer consisting of 64 neurons and *ReLU* activation, and train the model with *cross-entropy (CE)* loss objective for a total of 300 epochs at batch size 128, in all our experiments unless specified otherwise. We include additional experiments by changing training hyperparameters, i.e., batch size, learning rate, and model architecture, in Appendix E.4. We use train-val-test split of 0.7 : 0.1 : 0.2, and maintain the same split throughout all our experiments, i.e. we do not consider potential non-determinism introduced due to changing train-val-test splits. All our evaluations are performed on the test split. We will focus primarily on the ACSIncome dataset in the main text, while additional experiments on CelebA and ACSEmployment are included in the appendix.

**Fairness Metrics and Variance**  Fairness in machine learning has been interpreted widely into a multitude of definitions. In this paper, we will focus on average odds (AO), empirically interpreted as the average disparity between *true positive rates* and *false positive rates* of various groups. We also include additional results for equalized opportunity (EOpp) and disparate impact (DI) in Appendix F.3. For model predictions $R$, true labels $Y$, and sensitive attributes $A$, average odds can be defined as,

$$AverageOdds := \frac{1}{2} \sum_{y=\{0,1\}} |\mathbb{P}(R = 1|Y = y, A = 0) - \mathbb{P}(R = 1|Y = y, A = 1)|. \quad (1)$$

At the heart of our work is the study of fairness variance across model checkpoints. Unless otherwise specified, variance across multiple training runs refers to the variance across final checkpoints at epoch 300 of each training run. Similarly, variance across epochs refers to variance in a single training run across checkpoints at the end of every epoch for the last 200 epochs of training (i.e., from epoch 100 to epoch 300). We make this choice as the model has converged to stable accuracy scores before epoch 100 (refer to the training curve in Appendix A for more details).

**Non-Determinism in Model Training**  In this work, we focus on randomness due to stochasticity in the training algorithm, and we set manual seeds at various intermediate locations in our code to control the randomness. We refer to the seed set right before building the neural model as the weight

initialization seed, which influences the randomness in sampling the weight values. Similarly, we refer to the seed set right before the first training data shuffling as random reshuffling seed, which influences the data order that will be used as reference for the rest of the training. During training, we simply set the epoch number as seed right before reshuffling the reference data order at every epoch. As we can change the reference data order by changing the random reshuffling seed, this setup allows us to control non-determinism in data order throughout training with a single random seed. Fixing both weight initialization and random reshuffling seed allows us to deterministically replicate model training, while changing both seeds simultaneously is similar to the discussion of non-determinism present in literature. Pseudo code detailing the definition of various random seeds is provided in Appendix A. We also perform additional experiments on the impact of randomness introduced by dropout regularization in Appendix A.

## 3    THE SOURCES OF RANDOMNESS

We start by moving past the trivial observation that different training runs lead to different outcomes, and investigate the high fairness variance by studying the intricacies of the training dynamics.

### 3.1    WEIGHT INITIALIZATION AND RANDOM RESHUFFLING

Neural model training under widely adapted mini-batch stochastic gradient descent (SGD) has two main sources of non-determinism, i.e. weight initialization and random reshuffling. To decouple their impact on fairness variance, we perform the following experiment. We execute three sets of 50 unique training runs each while , *(i)* allowing for both sources of randomness, *(ii)* changing only the weight initialization while keeping the random reshuffling fixed, and *(iii)* changing only the random reshuffling while keeping the weight initialization fixed. For ACSIncome, we collect average odds (AO) and F1 score of all training runs in Figure 2 (see Figure 13 for the other datasets).

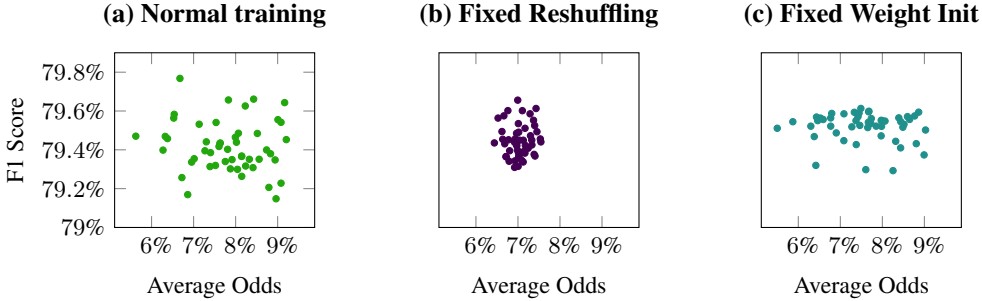

Figure 2: Average Odds (AO) variance by **(a)** allowing for both sources of randomness simultaneously, represents the instability in fairness scores as noted by existing literature. We see a significant drop in variance if we change only **(b)** the weight initialization while keeping the random reshuffling fixed. However, we observe high range of fairness variance by changing only **(c)** the random reshuffling, even for a fixed weight initialization. These results suggest random reshuffling of the data order as the dominant source of variance in fairness, with little influence from weight initialization.

The large range of fairness scores reported by allowing for both sources of randomness represents the variance observed in existing literature. Interestingly, when these sources are examined separately, the variance due to random reshuffling is equivalently large but the variance due to weight initialization drops significantly. It is clear that fairness variance originates from data order shuffling, while randomness in weight initialization has minimal impact. To further probe the difference between these two sources of non-determinism, we plot the median, inter-quartile range and overall range of average odds (AO) across the complete set of 50 training runs for the last 200 epochs in the two isolated settings from the previous experiment, i.e., changing only the weight initialization and only the random reshuffling, respectively. The resulting plots are collected in Figure 3.

We find high correlation in fairness scores across training runs with changing weight initialization (average pairwise pearson coefficient $\approx 0.91$), despite high variance between epochs as observed in Figure 3(a). This behavior supports our observations of low variance in fairness scores at the final

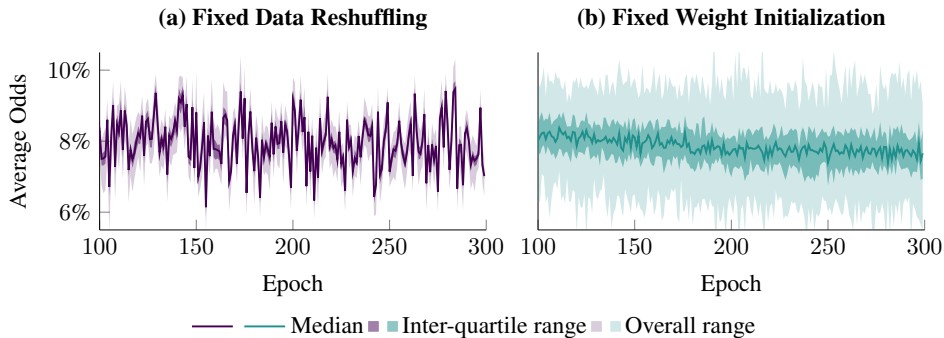

Figure 3: Median, inter-quartile range, and overall range of average odds across 50 different training runs while changing only the weight initialization or the random reshuffling respectively. **(a)** Changing only the weight initialization while keeping random reshuffling fixed implies that for any particular epoch, all training runs are fed the same data order. Despite different initializations, these models have very little variance across training runs, but high variance across epochs. This highlights the dominant impact of random reshuffling on model fairness. **(b)** Changing only the random reshuffling further supports our claim, as we see high variance even across training runs.

epoch in Figure 2. Furthermore, there is a lack of any reasonable correlation between fairness scores of training runs with changing random reshuffling (average pairwise pearson coefficient $\approx 0.13$), with noticeably high variance across both changing epochs and training runs in Figure 3(b). Interestingly, the high fairness variance across epochs inside a single training run in Figure 3(a) closely matches the variance that we observe across multiple training runs in Figure 3(b).

## 3.2 CHANGING PREDICTIONS AND DATA DISTRIBUTION

In previous experiments, we observed high variance in model fairness, and thus in turn it's predictive behavior, even between consecutive epochs. The changing predictive behavior of neural models is not surprising, and has been studied extensively in literature (Toneva et al., 2018; Kirkpatrick et al., 2017; Jagielski et al., 2022; Tirumala et al., 2022). As we are concerned with the fairness of the final decisions made by the model, we focus on a change in the model's discrete output class when discussing changing predictions. More specifically, a model is said to have undergone a change in prediction for some input $x$ during epoch $i$, if $\hat{y}_i(x) \neq \hat{y}_{i-1}(x)$, where $\hat{y}_i(x)$ is the output class when passing the input $x$ through the model checkpoint at the end of epoch $i$.

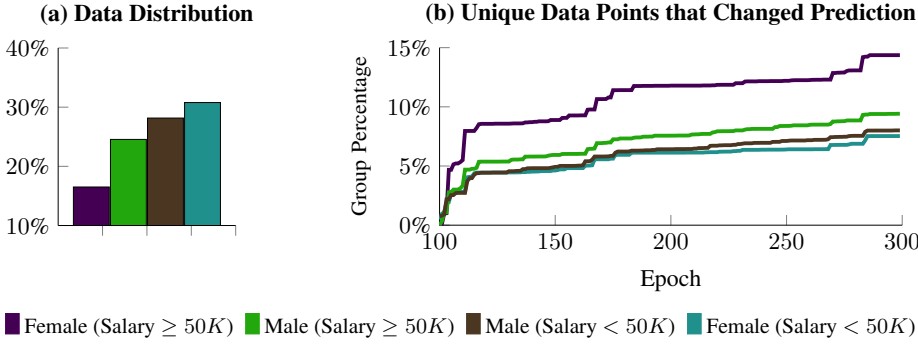

Figure 4: **(a)** The underlying data distribution of ACSIncome shows positive labels from group Female as an under-represented minority. **(b)** Total percentage of unique data points from each subgroup that change prediction across epochs follow the opposite order of their representation in the training data. These results highlight subgroups with least representation, i.e. positive labels from group Female in this setting, being the most vulnerable to changing predictions.

While these changing predictions maintain overall stable average performance, they can still have disparate impact on individual groups, the exact characteristics of which are less known. We study this instability by investigating individual data points which change their predictions. We plot the percentage of data points from each group that changed their prediction at least once between epochs 100 and $k$, where we gradually increase the value of $k$. The results are collected in Figure 4.

Clearly, the trends in percentage of unique data points with changing predictions mimics the distribution of each group in the original dataset, i.e. the groups which are represented the least are the most vulnerable to changing model behavior. In ACSIncome dataset, positive labels from group Female is severely under represented, and consequently can be seen to have almost twice the percentage of unique examples with changing predictions than any other group in the dataset. The higher instability of minority is an indication of higher uncertainty in the underlying model predictions. Thus, any fairness metric defined on the output of such a model will also reflect this instability, and manifests as variance in existing literature (Amir et al., 2021; Sellam et al., 2021; Soares et al., 2022).

## 4    DATA ORDER AND FAIRNESS

The previous set of experiments suggest the existence of an intricate relationship between the data order and resulting model fairness. In this section, we dig deeper into this relationship and exploit it to approximate fairness variance without separately training multiple models.

### 4.1    IMMEDIATE IMPACT OF DATA ORDER

A single epoch of SGD comprises of learning on a sequence of mini-batches from the training data, changing the model parameters one gradient update at a time. The data order during training governs the order of gradient updates and thus its impact on resulting model behavior is quite obvious (Bengio et al., 2009). However, our investigation into fairness variance suggests that this impact might be more immediate than expected, capable of stably predicting fairness scores irrespective of the model's preceding behavior. To test this, we design an experiment as follows. We sample 1000 unique checkpoints randomly from last 200 epochs of 50 different training runs while allowing both forms of training non-determinism simultaneously. These checkpoints represent a variety of models with different weight initialization, training data order and even number of training epochs. We then train each of these checkpoints for exactly one epoch on a common randomly chosen data order. We collect fairness variance across checkpoints before and after this single epoch of training in Figure 5.

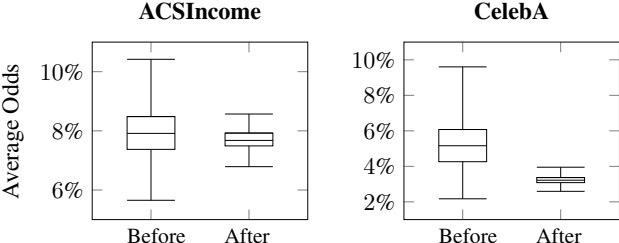

Figure 5: (*Before*) Multiple checkpoints taken from training runs with changing weight initialization, training data order, and even number of training epochs, show a high range of fairness variance (i.e., average odds variance), as expected. (*After*) However, these fairness scores are stabilized significantly by training these checkpoints for only a single epoch on a common randomly chosen data order. This shows the immediate impact of data order on model fairness.

Before training on the same data order for a single epoch, these checkpoints represent the complete range of fairness variance previously noted. However, after only a single epoch of training on a common data order, these models have all moved towards the same fairness score with significantly lower variance. This highlights the immediate impact of data order of model fairness, which is stable based on only the data order of the most recent epoch. An existing line of research suggests the presence of no energy valleys in deep learning loss landscape between minimas of separately trained models (Draxler et al., 2018; Garipov et al., 2018), which could explain the immediate impact of data order, as various checkpoints might be connected in the loss landscape. Another possible explanation

of models converging to the same fairness scores relies on a recent work that shows that there is only one functionally unique minima in loss landscape of neural models (Ainsworth et al., 2022). Thus, it is possible that different models when fed the same data order for even one epoch move towards a functionally common model, with possible permutation symmetries in them.

Epochs are pseudo constructs, used to record multiple passes through the data, and thus we expect the influence of data order on model fairness to have a granular manifestation as the few most recent gradient updates of the model. Furthermore, while we have shown that the fairness variance is indeed reduced, we are yet to show that it is also predictable. To achieve both simultaneously, we extend the experiment above as follows. We start by extracting the data order corresponding to the epochs achieving best and worst fairness scores on the validation set in a standard training run. Next, we train these checkpoints on exactly $b$ batches, taken from the suffix of data order with best and worst fairness scores as described above. This allows us to obtain models which have experienced the same most recent $b$ gradient updates. For example, at $b = 0$, this setup involves no training and we simply report the variance across the original set of checkpoints, while at $b = 1070$ (i.e., total number of batches in an epoch for ACSIncome), this setup will repeat the previous experiment. We test for $b = \{0, 20, 50, 100, 200\}$ and the results are collected in Figure 6.

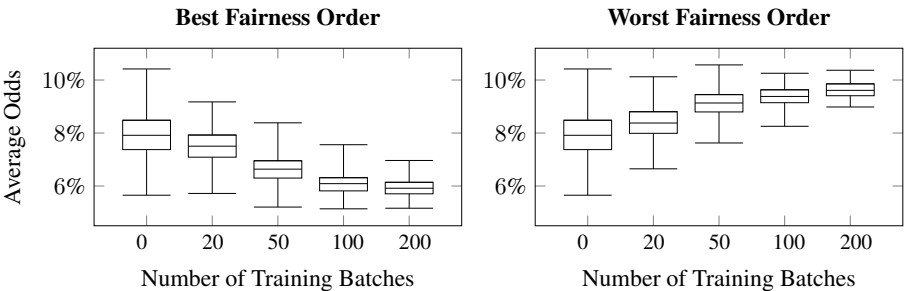

Figure 6: Fairness scores (average odds) stabilize as the number of most recent common training batches increase, highlighting the impact of these gradient updates on model fairness. Moreover, it stabilizes to low or high fairness scores based on the corresponding data order from the epoch with **best** or **worst** fairness score, showing the predictability of model fairness for any given data order.

As the number of fixed batches $b$ increases, the model fairness gets more stable, and already at $b = 200$, the model fairness is as stable as seen in previous set of experiments. This shows that it is indeed the most recent gradient updates seen by the model which dominate its fairness scores, irrespective of model behavior prior to those updates. Moreover, the trends of stability move towards expected fairness based on the choice of original data order taken from best or worst fairness epoch as described earlier, which helps us show the predictability of the impact of data order.

## 4.2 Capturing Fairness Variance in a Single Training Run

We now return to our original problem of capturing fairness variance without wasting computing resources on a large number of training runs. We saw stability and predictability in fairness scores based on the data order, and thus fairness variance across multiple training runs is simply the randomness in data order across their last epochs. As these data orders are obtained by random shuffling, their distribution across multiple training runs should result in the same empirical range of fairness as their distribution across epochs. Thus, we propose evaluating intermediate checkpoints in a single training run as a proxy for reporting fairness variance across multiple training runs.

We test this by first plotting the distribution of fairness scores for sampling across multiple training runs (three different stopping epochs) and sampling across epochs (three different training runs) in Figure 7(a). Furthermore, we also test the quality of black swans, i.e. the best models captured, as a function of number of unique training runs and number of epochs evaluated per training run. For a data point $epochs = e; seeds = s$, we perform $s$ unique training runs, and evaluate the model for last $e$ epochs in each training run, thus accumulating a total of $e * s$ checkpoints. These checkpoints are then used to calculate some quality measure, and averaged over 50 repeats to compensate for randomness in choosing $s$ training runs. The experiment is performed for all combinations of $e \in [1, 50]$ and $s \in [1, 50]$. We use two different quality measures for a set of checkpoints as described above, *(i)* the

best fairness achieved across all checkpoints, and *(ii)* the Hausdorff distance (Birsan & Tiba, 2005) of the pareto-front (including both fairness and F1 scores) from the best achievable pareto-front, i.e., for $e = 50; s = 50$. The results are collected in Figure 7(b).

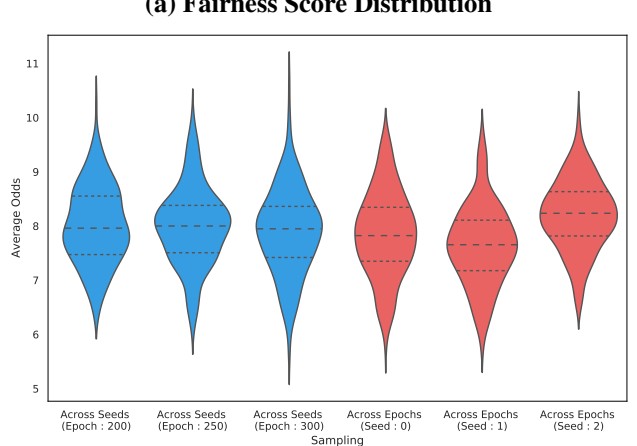
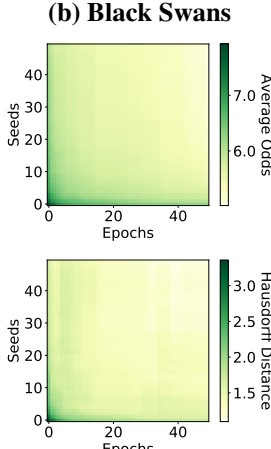

Figure 7: **(a)** Fairness scores (average odds) across multiple training runs and across epochs in a single training run have similar empirical distributions. Thus, studying this distribution across epochs provides a highly efficient alternative to executing multiple training runs. **(b)** Quality of black swans (i.e. extremely rare checkpoints) improves with more checkpoints collected, either in terms of fairness (the lowest achievable average odds score) or the trade-off between overall performance and fairness (the hausdorff distance to the best pareto front). This improvement occurs at the same rate, irrespective of sampling checkpoints across multiple random seeds (x-axis) or multiple epochs in a single seed (y-axis). But, sampling across epochs is significantly cheaper.

As we can notice from the plot in Figure 7(a), the empirical distribution of fairness across multiple training runs closely matches the distribution across a single training run. We even performed the Kolmogorov–Smirnov (KS) test (Massey Jr, 1951) to match sampling across multiple runs (at epoch 300) and sampling across epochs (for a single training run with seed 0) and found the maximum difference in empirical CDF to be only $0.07$, with a p-value of $0.712$, i.e. the probability that both set of checkpoints were sampled from the same underlying distribution. Moreover, the quality of black swans in Figure 7(b) for both quality measures suggests similar trends. For example, when comparing the two extremes, i.e. $e = 1; s = 50$ and $e = 50; s = 1$, we see similar results, with latter requiring $50$ times less computation. This shows that the commonly used method to capture fairness variance in literature (i.e., $e = 1; s = 50$) is highly inefficient use of computing resources, and one can achieve the same by simply observing model behavior across a single training run. With these experiments, we show direct benefits of evaluating multiple epochs in a single training run, saving huge amounts of resources and time in capturing the overall model variance.

## 5  MANIPULATING GROUP LEVEL ACCURACY WITH DATA ORDER

In previous sections, we explored the instability in model fairness and took advantage of our insights to reduce computational requirements for studying this variance. However, the predictability of fairness scores based on the most recent batches implies the possibility that we should be able to finely manipulate the fairness scores, or more precisely the group level accuracy, by simply controlling the data order. We hypothesize that the data distribution seen by the model in the most recent batches can temporarily change the loss landscape and nudge the group level accuracy behavior towards the chosen distribution. Coupled with the immediate impact of data order shown in previous sections, this could allow us to manipulate group level accuracy in only a single epoch of fine-tuning.

To test this hypothesis, we perform the following experiment. We start by defining the ratio between different groups, to indicate the distribution that will dominate the model behavior. For example, a ratio of $1 : 10$ between two groups will distinctly favor the latter group, while a ratio of $10 : 1$ will favor the first group. We then form batches in the exact ratio as provided until we run out of data

points (which will happen for any ratio that is not the exact distribution of the original dataset). The excess data points are then shuffled randomly and placed at the prefix of data order.

Similar to previous experiment settings, we pick the last epoch checkpoint from 50 unique training runs while allowing for both forms of non-determinism (i.e. weight initialization and random reshuffling), and then train them all for a single epoch on our custom data order. For our first experiment, we fix all other ratios and only change the ratio between positive and negative labels in group Female. We record the group level accuracy for all subgroups separately, along with the overall model accuracy. We repeat the experiment by now changing only the ratio between positive and negative labels in group Male. The results are collected in Figure 8.

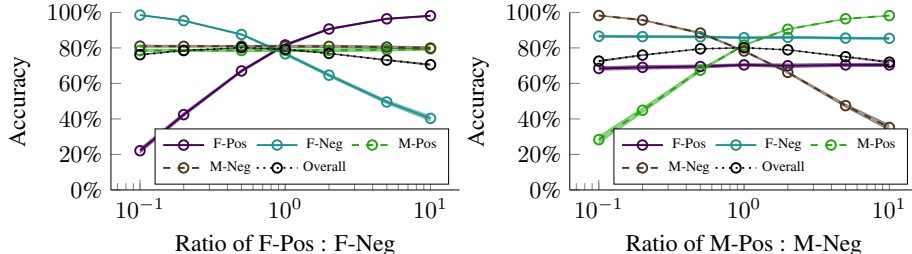

Figure 8: We show how to control group level accuracy by changing data distribution of the most recent gradient updates, tested separately for (a) ratio between positive and negative labels for group Female, and (b) ratio between positive and negative labels for group Male, while keeping other ratios fixed. In only a single epoch of training, we are able to manipulate the group level accuracy trade-off, with relatively small impact on overall accuracy.

It is clear that by manipulating the data distribution seen by the model in the most recent gradient updates, we can control the group level accuracy of the model. While the overall accuracy drops noticeably for extreme ratios, it does not change much in the middle despite significant variance in group level performances. This further strengthens our claim on the dominance of data order on model fairness and group level accuracy. *Fairbatch* (Roh et al., 2020), a state-of-the-art bias mitigation algorithm, follows a similar formulation as our custom data order. They create batches with fixed ratio between groups, and the ratio is optimized at every epoch to counter the existing bias in the model. Our results not only explain the success of their algorithm, but also state that instead of regularly adapting the distribution to compensate for the model bias, one can directly use the final desired distribution and the model will adapt to it immediately.

Finally, we extract two special cases from the above experiments, i.e. ratios of $1:1$ and $1:3$ between positive and negative labels of group Female (for ACSIncome dataset), and call them *EqualOrder* and *AdvOrder* respectively. While *EqualOrder* achieves competitive performances with state-of-the-art bias mitigation methods, *AdvOrder* is capable of derailing the fairness scores of various mitigation algorithms. Note that we can force even worse fairness gaps than achieved by *AdvOrder* by pushing the ratio to its extreme, however that will also noticeably impact the model's overall accuracy. We perform experiments with three unique setups, standard training and two commonly used bias mitigation algorithms. This includes, (i) *Reweighing* (Kamiran & Calders, 2012), a data pre-processing algorithm which weighs every label-group pair based on its representation in the overall dataset, and (ii) *Equalized Odds Loss* (Fukuchi et al., 2020), an in-processing loss function to nudge the model towards fair predictions. We incorporate both *EqualOrder* and *AdvOrder* in these setups to see the influence of data order. The results are collected in Figure 9.

By simply training with *EqualOrder* for a single epoch, the baseline models achieve competitive fairness scores to state-of-the-art bias mitigation algorithms. *Reweighing* suffers from an unexpected high bias under *EqualOrder*, as the combination of ideally distributed data order along with increase in the weight of minority data points pushes the model towards significant unfair behavior against the majority. On the other hand, using *AdvOrder* in the baseline setting can further push the model bias, emphasizing the adversarial power of data ordering. Both *Reweighing* and *Equalized Odds Loss* also suffers similar fates, but with controlled damage in latter due to the loss function regularly adapting to the degrading behavior. These results together cement the effectiveness of manipulating group level accuracy by controlling the data order for just a single epoch of training.

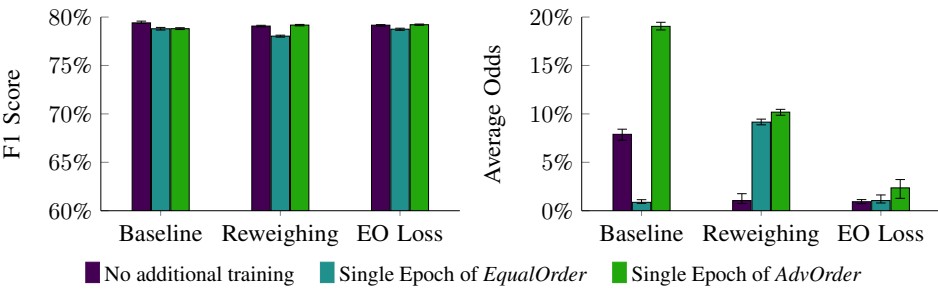

Figure 9: Under baseline training setup, by changing only the data order for just a single epoch of training, *EqualOrder* gets competitive performance to commonly used bias mitigation methods. Similarly, *AdvOrder* gets significantly worse fairness than even the baseline. For Reweighing, we see an increase in bias even with *EqualOrder*, as the beneficial weighing of minority creates bias towards the majority. On the other hand, even reweighing is not enough to counter the effects of *AdvOrder*. Finally, equalized odds loss is capable of dynamically adapting to the model's changing predictive behavior, yet we still observe noticeable increase in bias under *AdvOrder*.

## 6 RELATED WORK

**Variance in Fair Machine Learning**    Recently, there has been a growing awareness of high variance and instability in fair deep learning, associated with non-determinism in model training or underlying implementations (Amir et al., 2021; Sellam et al., 2021; Soares et al., 2022; Qian et al., 2021), and the challenges it creates in trusting existing results. The high variance in fairness scores has even been used to propose a new leave-one-out unfairness metric (Black & Fredrikson, 2021).

Soares et al. (2022) refute various trends in literature associated with fairness of large language models by simply executing multiple identical training runs, while Sellam et al. (2021) focus on the impact of randomness in pre-training paradigm and find further increase in instability. Amir et al. (2021) even revisit state-of-the-art bias mitigation in clinical texts and show a lack of statistically significant improvement after accounting for non-determinism in training. While existing literature focuses on exploring the impact of fairness variance, we instead focus on investigating the source of this non-determinism. Furthermore, we propose to move away from the practice of executing multiple training runs to capture this variance and instead provide a computationally efficient proxy.

**Data Order and Example Forgetting**    Controlling the data order during training has gained noticeable success in improving convergence speeds (Bengio et al., 2009; Shah et al., 2020; Soviany et al., 2022; Mohtashami et al., 2022), while its adversarial capabilities have also been explored (Shumailov et al., 2021). Even under randomly shuffled data order, neural models are known to undergo changes in predictive behavior during training (Toneva et al., 2018; Kirkpatrick et al., 2017; Jagielski et al., 2022; Tirumala et al., 2022). We extend the discussion on non-determinism in model behavior to model fairness, and investigate its connections with data order. We focus specifically on the instability in predictions made by the learning model and not its internal state, as we are concerned about the disparity in group level decisions which highlights the encoded bias.

## 7 CONCLUSION

Fairness variance in deep learning has raised concerns regarding the reliability of existing results in literature. In this paper, we took a closer look at this variance and proposed a proxy to training multiple models by studying fairness scores inside a single training run, which exploits a surprisingly dominant impact of data order on model behavior. We even manipulated the data order to minutely control group level accuracy without losing overall accuracy in only a single epoch on training. In our work, we focused only on the discrete decisions made by the model, as we were investigating the impact of non-determinism in model training on its fairness. However, further extensions of this discussion to trends in the internal state of the learned model can reveal even granular characteristics, and has potential application in similar fields of research suffering from high variance.

## 8 REPRODUCIBILITY STATEMENT

Our work studies various sources of non-determinism in neural model training and its impact on model fairness. Thus, it is important to address the implementation choices we have made to allow easy reproducibility. We provide a minimal psuedo code in Appendix A to clearly define various random seeds used in our setup to control training non-determinism. The complete training and evaluation pipeline is deterministic outside the choice of these random seeds, i.e., one can perfectly replicate the training by simply fixing these seeds. We also provide our code to train models and perform several evaluations done in the paper as supplementary material to allow easy adaptation for future research.

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

# Appendix

## Table of Contents

## A    NON-DETERMINISM IN MODEL TRAINING

We provide a minimal pseudo-code to explain our control over training non-determinism and define weight initialization and random reshuffling seeds.

```
....
torch.manual_seed(weight_initialization_seed)
model = MLPModel() # Initialize Neural Model
...
train_data, .. = load_dataset() # load data with deterministic order
train_data = shuffle(train_data, seed=random_reshuffling_seed)
...
trainloader = torch.utils.data.DataLoader(train_data, ..., shuffle=True)
...
for epoch in range(total_epochs):
    ...
    torch.manual_seed(epoch)
    for batch in trainloader:
        ...
```

## B    DATASETS

**ACSIncome**    ACSIncome is one of the five pre-defined tasks in the Folktables dataset Ding et al. (2021), which was recently collected to improve the older and commonly used UCI Adult Income dataset Dua & Graff (2017). More specifically, we use the subset of Folktables dataset from the state of California, USA for 2018. The dataset contains a total of $195,665$ data points with $10$ features each, where each data points represents an individual. The task is a binary classification to predict whether the individual's income is above $50,000$. For fairness measures, we use gender (Female vs Male) as the sensitive attribute, which is also one of the $10$ input features.

List of features : *Age*, *Class of worker*, *Educational attainment*, *Marital status*, *Occupation*, *Place of birth*, *Relationship*, *Usual hours worked per week in past 12 months*, *Sex*, *Recoded detailed race code*

**ACSEmployment**    ACSEmployment is another one of the five pre-defined tasks in the Folktables dataset Ding et al. (2021). We use the same subset from the state of California, USA for 2018 as above. The dataset contains a total of $378,817$ data points with $16$ features each, and the task is a binary classification to predict whether the individual is employed or not. For fairness measures, same as above, we use gender (Female vs Male) as the sensitive attribute.

List of features : *Age*, *Educational attainment*, *Marital status*, *Sex*, *Disability recode*, *Employment status of parents*, *Mobility status*, *Citizenship status*, *Military service*, *Ancestry recode*, *Nativity*, *Relationship*, *Hearing difficulty*, *Vision difficulty*, *Cognitive difficulty*, *Recoded detailed race code*, *Grandparents living with grandchildren*

**CelebA**    CelebA dataset is a large scale celebrity face attributes dataset Liu et al. (2015), which contains a total of $202,599$ images of celebrities with $40$ different binary labels each. We focus on the binary classification task of the 'smiling' label in the dataset, while we use the 'gender' label as the sensitive attribute for fairness evaluation. Moreover, we do not directly use the images of CelebA dataset, but instead pass them through a pre-trained and frozen ResNet-50 backbone He et al. (2016) to extract image representations, which are treated as inputs to our model.

## C  TRAINING CURVE AND CONVERGENCE

To understand why we choose to study the model behavior between epochs 100 and 300, we point the reader towards the overall training curve of the model plotted in Figure 10. It is clear that the model has converged by epoch 100, and maintains stable accuracy scores for the last 200 epochs.

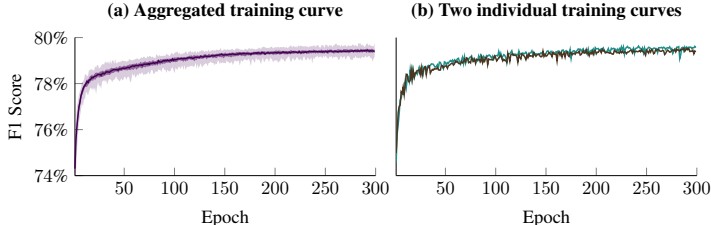

Figure 10: (Left) Aggregated training curve for all 50 training runs with both changing weight initialization and random reshuffling. (Right) Example of two individual training runs.

Even though the accuracy scores have converged, we still find high variance in fairness scores as discussed in the main text of the paper. One might suspect this implies that fairness scores could take longer to converge. To check this, we allow a single training to run for a total of 3000 epochs (as opposed to the standard 300 epochs used in all other experiments in our paper) and collect the fairness scores in Figure 11. It is clear that the model fairness does not stabilize by simply increasing the number of epochs, which further supports our hypothesis of never-ending local oscillations due to SGD noise that cause high fairness variance.

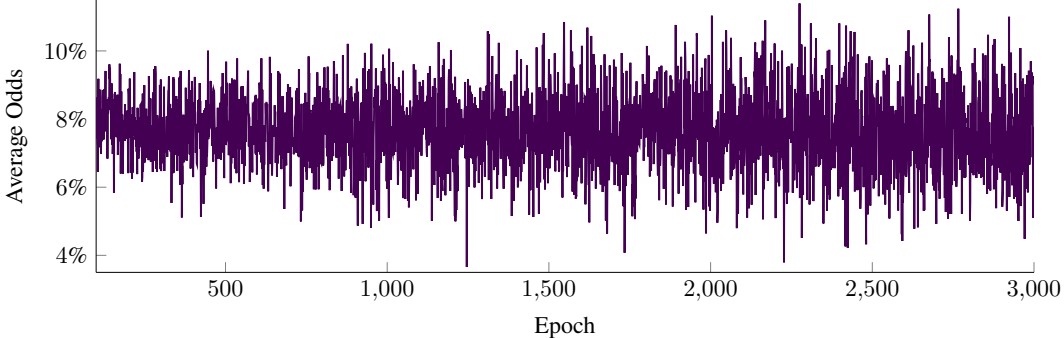

Figure 11: A single training run extended to 3000 epochs.

# D DROPOUT REGULARIZATION

Another notable source of randomness in neural model training is dropout regularization (Srivastava et al., 2014). Dropout regularization randomly drops a certain percentage (known as dropout rate) of connections between consecutive layers in the model at every forward pass during training. We extend our discussion from Section 3 to study the impact of dropout on the trends of random reshuffling seen in Figure 3. More specifically, we repeat the experiments while introducing various rates of dropout in the training setup. The results are collected in Figure 12.

Randomness in dropout regularization under fixed weight initialization and fixed random reshuffling (Figure12(e), (h)) introduces minute variance across multiple runs. While it adds noise to previously noted trends, the variance in fairness across consecutive epochs in a single training run, as well as the dominance of random reshuffling on fairness variance is still clearly noticeable (Figure 12(c), (d), (f), (g)). Interestingly, the overall range of fairness variance does go down with higher dropout rate, but gradually impacts the final achieved accuracy scores (Figure 12(j), (k)) as well as fairness.

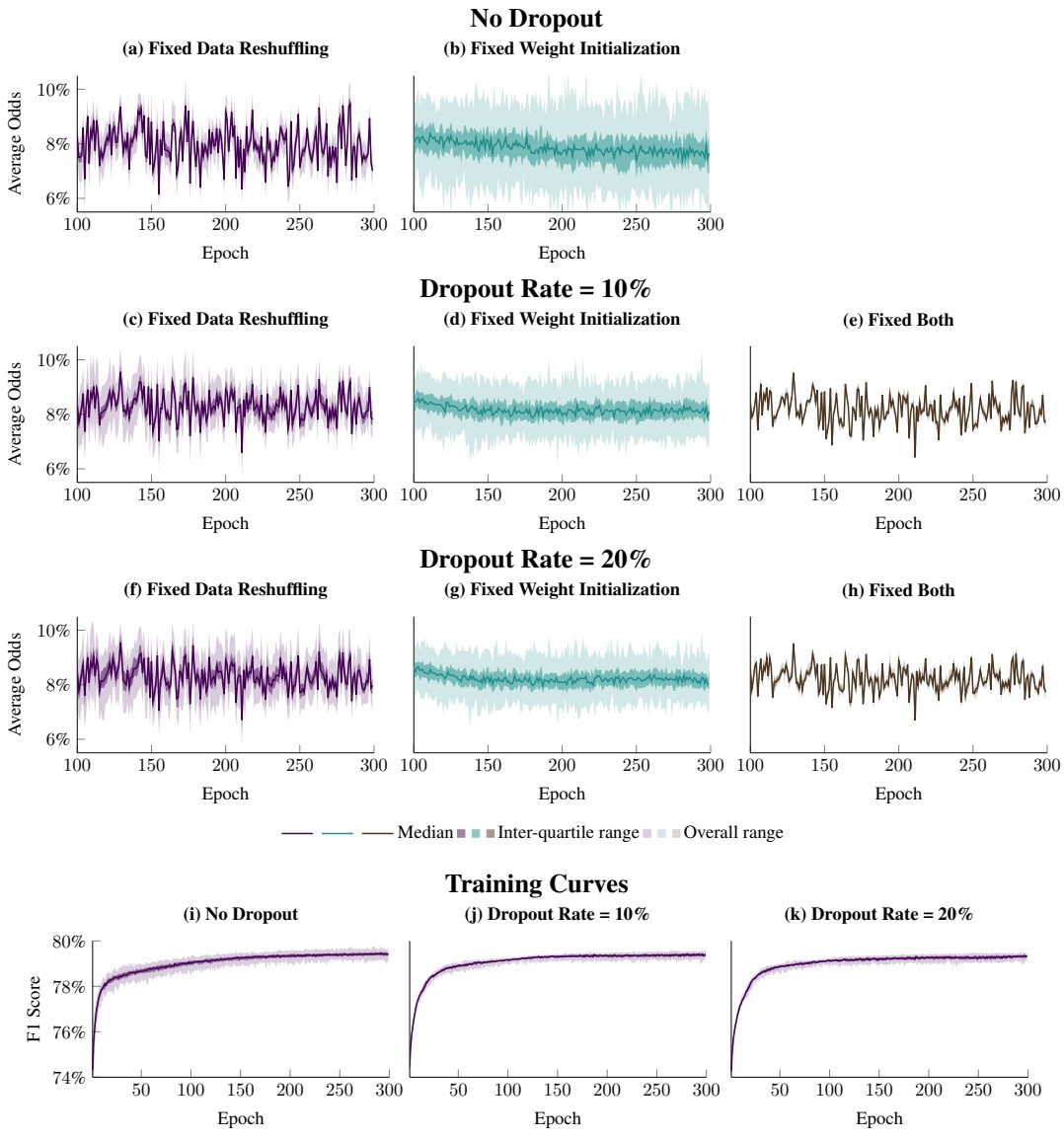

Figure 12: Additional experiments with dropout regularization reveal similar trends as in Figure 3. These results further highlight the dominant impact of random reshuffling on fairness.

# E  ADDITIONAL EXPERIMENTS FOR ALL DATASETS

## E.1  WEIGHT INITIALIZATION AND RANDOM RESHUFFLING

We provide additional results in Figure 13, 14 on CelebA and ACSEmployment dataset to show the dominance of random reshuffling on model fairness.

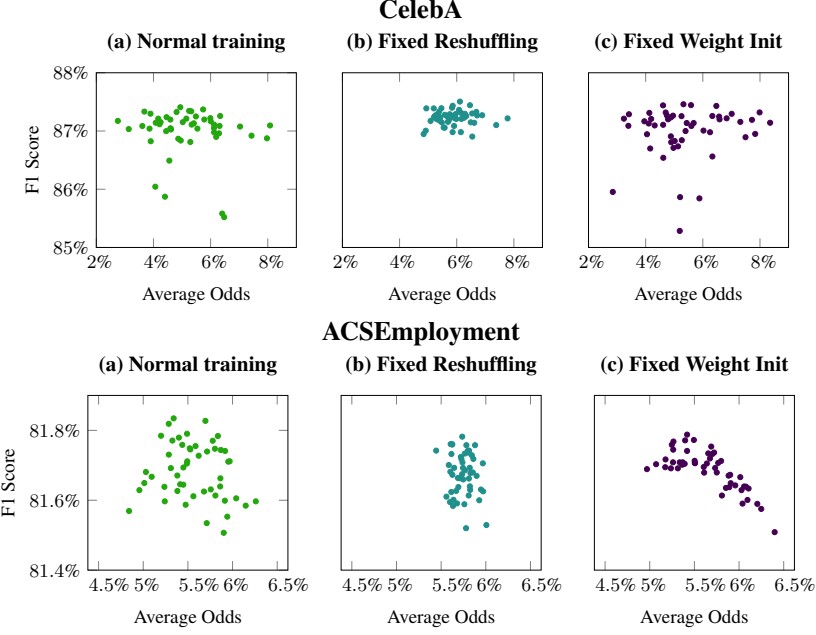

Figure 13: Additional experiments on CelebA and ACSEmployment datasets reveal similar trends as seen in Figure 2. Random reshuffling of data order is the dominant source of variance in fairness scores, with very little influence from the weight initialization.

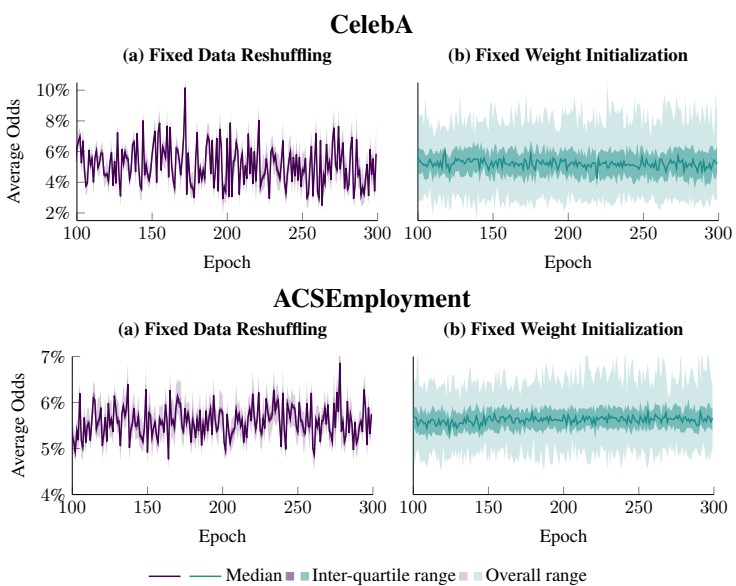

Figure 14: Additional experiments on CelebA and ACSEmployment datasets reveal similar trends as in Figure 3. These results further highlight the dominant impact of random reshuffling on fairness.

### E.2 CHANGING PREDICTIONS AND DATA DISTRIBUTION

We show the relationship between data distribution and changing predictions for CelebA and ACSEmployment in Figure 15. The group with least representation maintains being the most vulnerable to changing predictions.

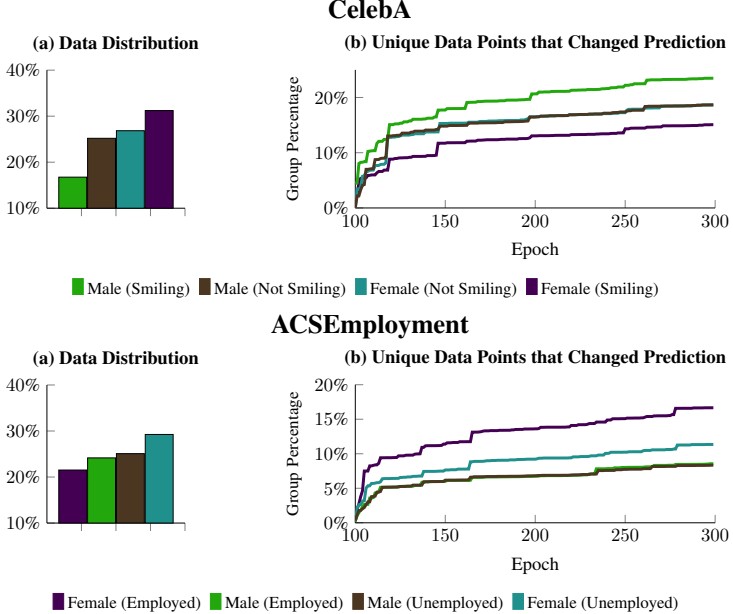

Figure 15: Additional experiments on CelebA and ACSEmployment datasets reveal similar trends as seen in Figure 4. These results highlight subgroups with least representation being the most vulnerable to changing predictions.

We showed that the fairness scores are dominated by the most recent gradient updates as seen by the model. As a sanity check, we also provide ablation study for experiments in Figure 6, but choose the $b$ batches randomly instead of from the prefix. The results are collected in Figure 16. The results show that it is indeed the batches from data order prefix that govern the predictability of model fairness.

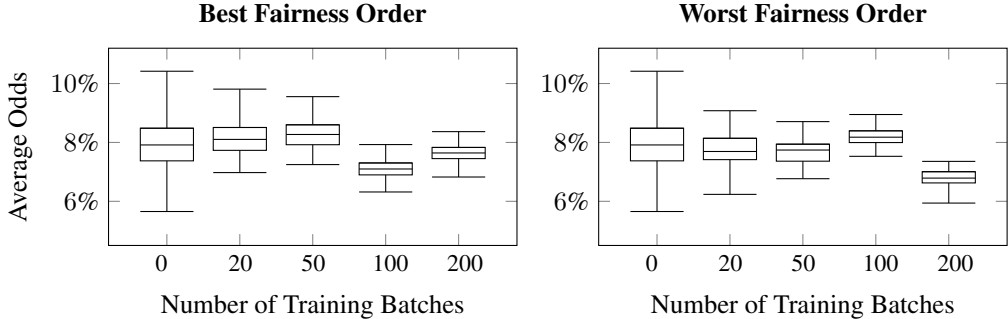

Figure 16: Additional experiments by choosing $b$ batches, similar to the experiments in Figure 6, but choosing batches randomly instead of the prefix. The results still maintain stability as $b$ increases but now they stabilize to random fairness scores instead of best and worst fairness scores as seen in Figure 6. Clearly, its the most recent batches in that order which truly governs the model fairness.

## E.3 CAPTURING VARIANCE IN A SINGLE TRAINING RUN

We show the empirical similarity of distribution across multiple training runs and multiple epochs in a single training run for additional datasets CelebA and ACSEmployment in Figure 17.

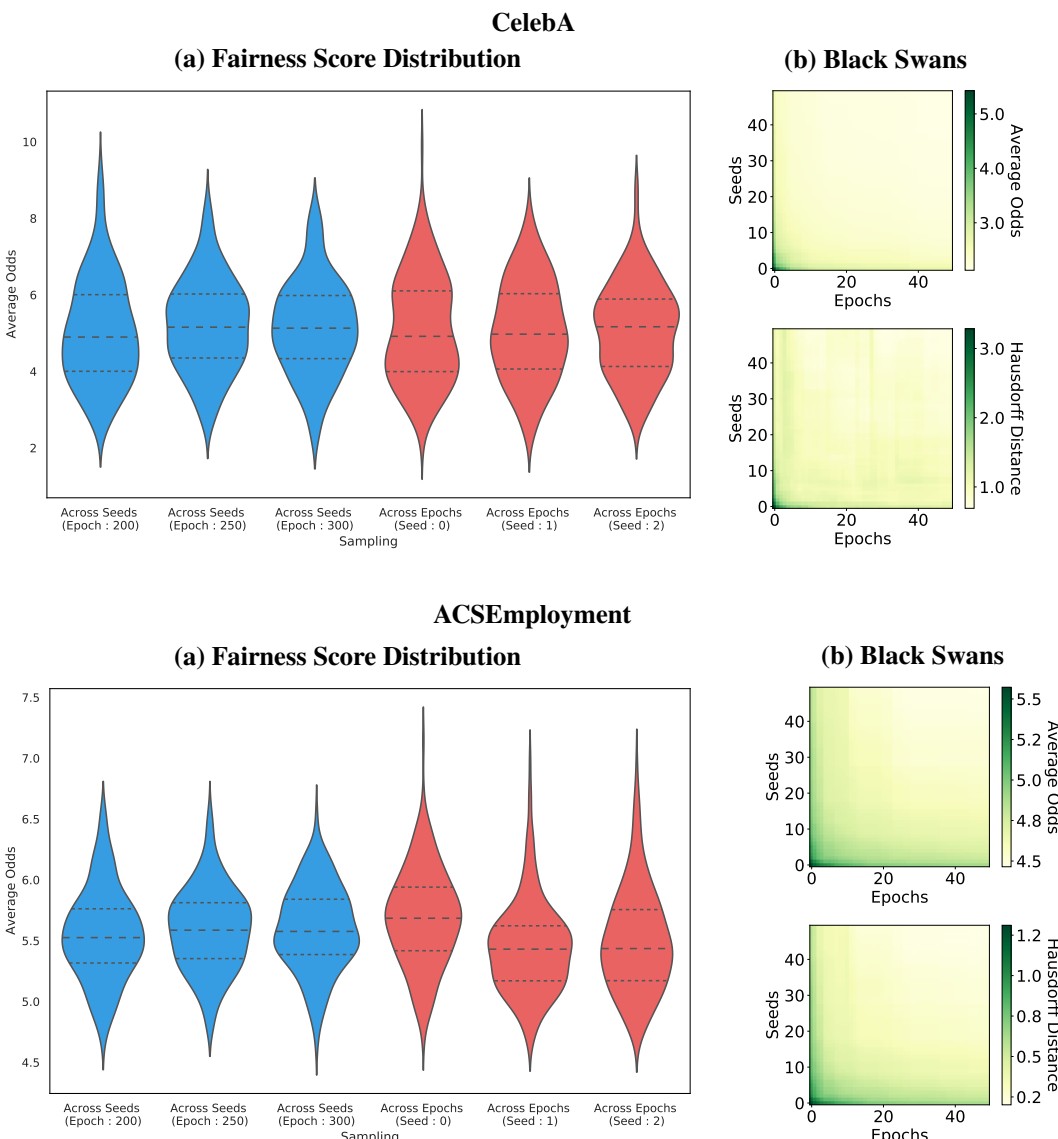

Figure 17: Additional experiments on CelebA and ACSEmployment datasets reveal similar trends as seen in Figure 7. Fairness scores (average odds) across multiple training runs and across epochs in a single training run have similar empirical distributions. Thus, studying this distribution across epochs provides a highly efficient alternative to executing multiple training runs.

## E.4 Manipulating Group Level Accuracy with Data Order

We provide additional results in Figure 18, 19, highlighting the predictability of model fairness based on the data order, and its impact compared against various state-of-the-art baselines.

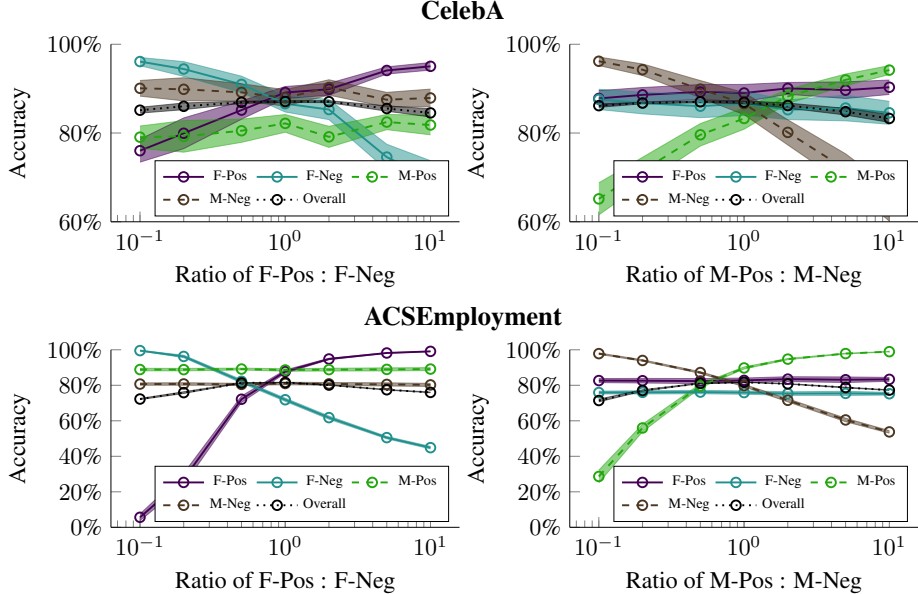

Figure 18: Additional experiments on CelebA and ACSEmployment datasets reveal similar trends as seen in Figure 8. In only a single epoch of training, we are able to manipulate the group level accuracy trade-off, with relatively small impact on overall accuracy.

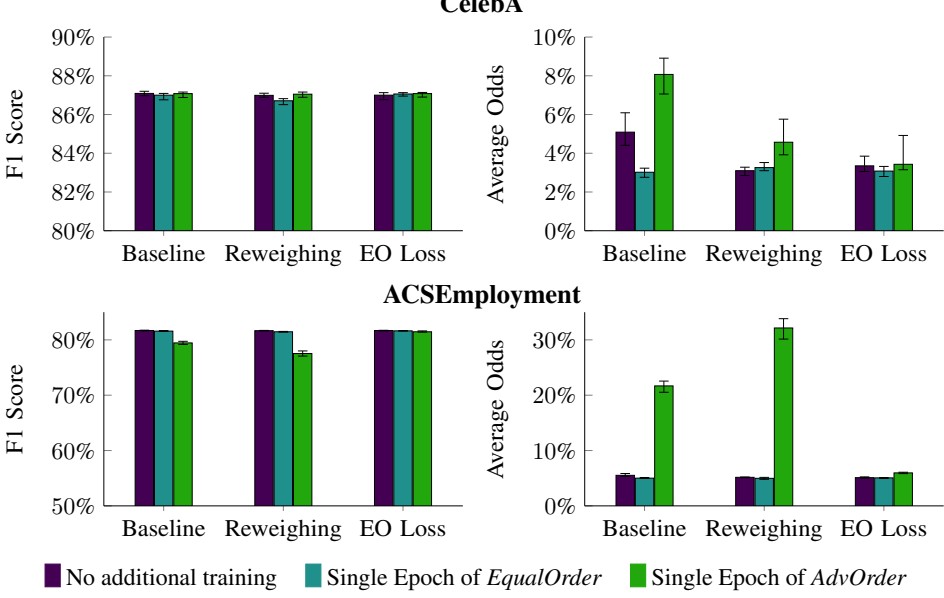

Figure 19: Additional experiments on CelebA and ACSEmployment datasets reveal similar trends as seen in Figure 9. *EqualOrder* gets competitive performance to commonly used bias mitigation methods. Similarly, *AdvOrder* gets significantly worse fairness than even the baseline.

# F  ADDITIONAL EXPERIMENTS FOR CHANGING HYPERPARAMETERS

We provide additional experiments for *(i)* batch size 16 and 1024, deviating from the default batch size of 128 in the main text, *(ii)* learning rate 0.01 and 0.0001, deviating from the default learning rate of 0.001 in the main text, and *(iii)* model architecture with two hidden layers containing 2048 and 64 neurons respectively, deviating from the single hidden layer architecture used in the main text.

## F.1  WEIGHT INITIALIZATION AND RANDOM RESHUFFLING

We provide additional results in Figure 21 for changing batch size, learning rate, and architecture on ACSIncome dataset to show the dominance of random reshuffling on model fairness. With a decrease in batch size, the range of fairness variance increases significantly, but the overall expected trends follow the same behavior as noted in the main text, i.e. a sharp change in fairness scores across epochs under fixed data reshuffling, and high variance even for a single epoch across multiple runs under changing data reshuffling (fixed weight initialization). Similar trends can be observed when we increase the learning rate, or provide the training algorithm with a bigger neural model. The increase in variance suggests high instability with smaller batch size, higher learning rate, and bigger neural models, all of which is expected.

On the other hand, one would expect more stable model behavior with a bigger batch size or a smaller learning rate. While this is indeed the case, the comparison here is not fair because these models have not yet converged, as evident by the clear downward trend of fairness scores in Figure 21. Note that compared to these models, the original set of models converged before epoch 100 (see Appendix A). To further highlight the same, we also show the training curve of all 5 settings compared against the default setup in Figure 20. It is clear that certain hyperparameter settings are not conducive to efficient convergence, even though they might eventually provide more stable fairness scores.

Finally, to understand the difficulty of reaching convergence, we continue training a single instance with batch size 1024 (and learning rate 0.0001) for a total of 1000 epochs and compare it against the standard training setup (i.e., with batch size 128 and learning rate 0.001). The results are collected in Figure 22, 23. It is clear that a larger batch size (or lower learning rate) significantly slows down the convergence speed. Moreover, it also does not achieve the same accuracy scores previously seen, possibly due to not being able to take complete advantage of the noise usually found in mini-batch gradient descent. Thus, using a high number of training epochs for a model with hyperparameters that enforce stability (i.e. large batch size or lower learning rate) is an inefficient solution to solving the problem of fairness variance.

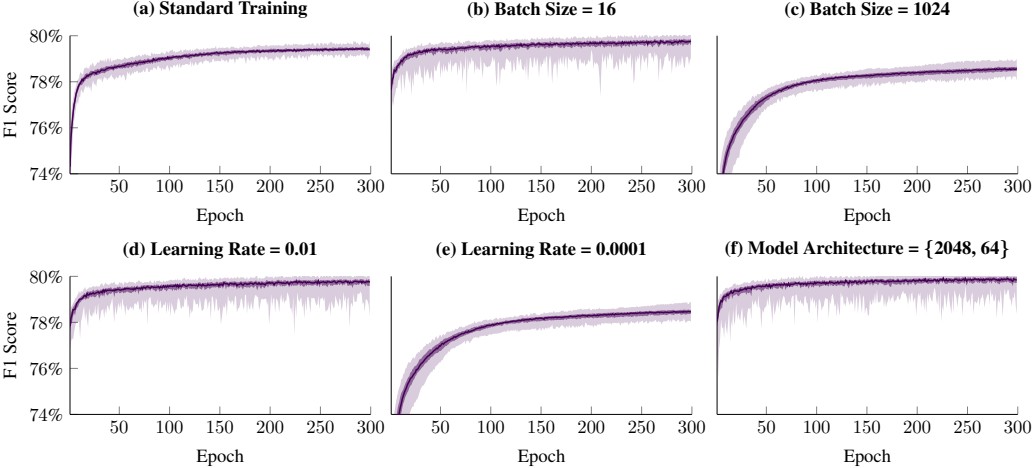

Figure 20: Training curve of the five new settings, along with original training curve copied here for easy reference. It is clear that decreasing the batch size, increasing learning rate, or using a bigger model architecture results in a faster model convergence but with higher variance even in accuracy scores. On the other hand, models with high batch size or low learning rate tend to not achieve the same accuracy scores and are still improving even at 300 epochs.

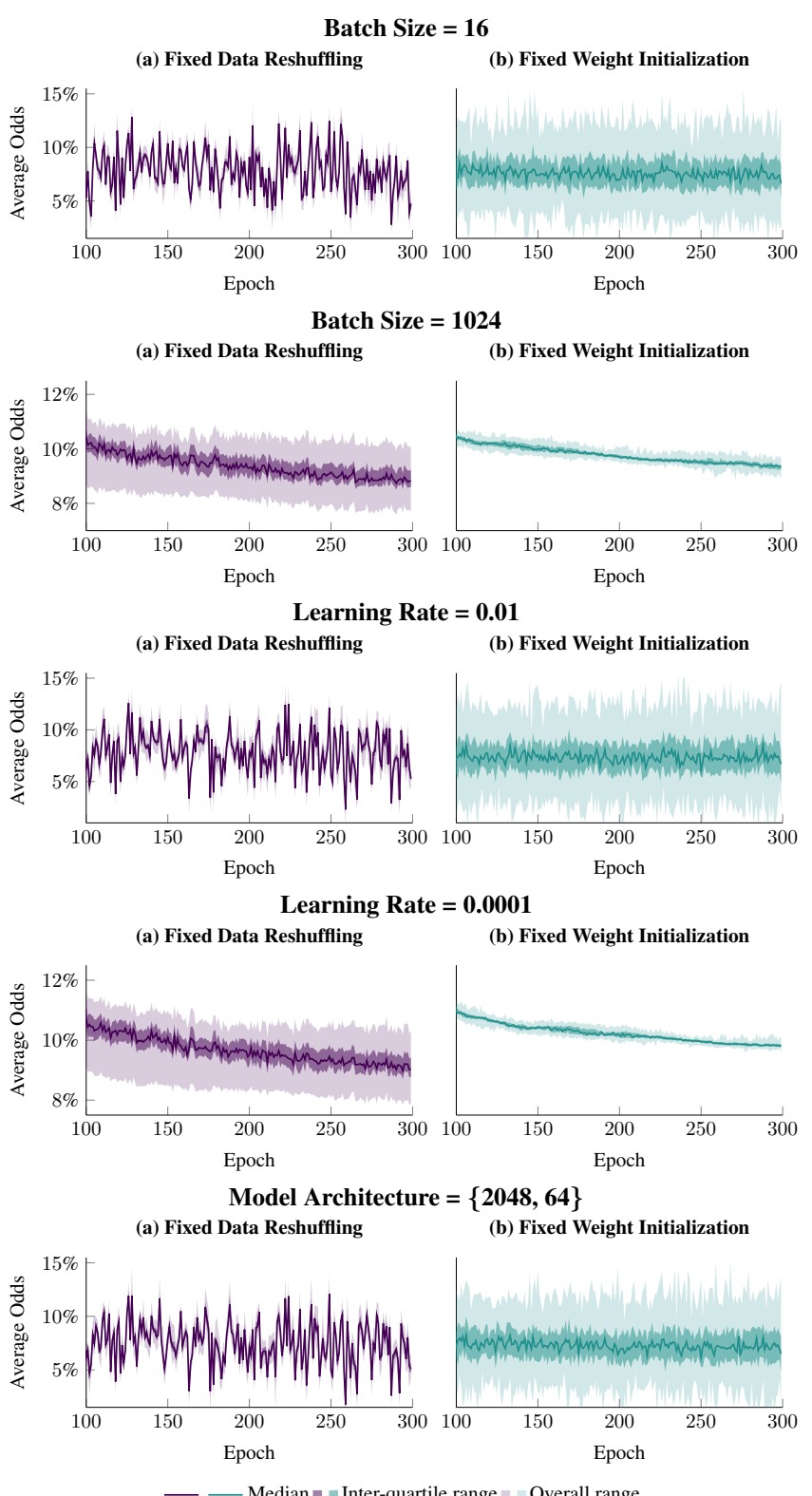

Figure 21: Additional experiments for changing training hyperparameters with experiment setting as in Figure 3. These results further highlight the dominant impact of random reshuffling on fairness.

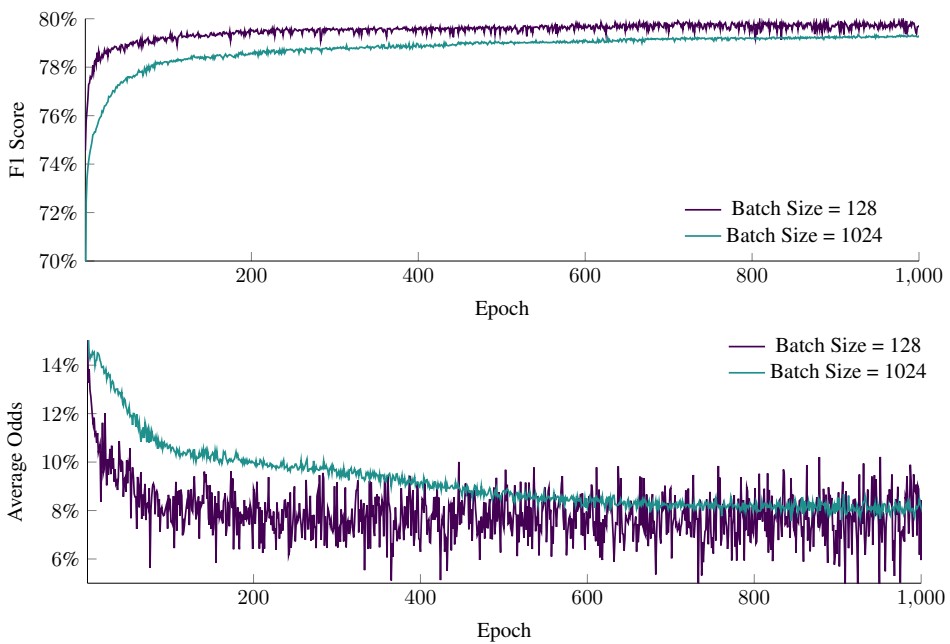

Figure 22: Using a higher batch size can over time achieve reasonably stable fairness scores, however the convergence speed is significantly slower. Moreover, the higher batch size does not convergence to the same accuracy scores, and looses a noticeable margin of accuracy.

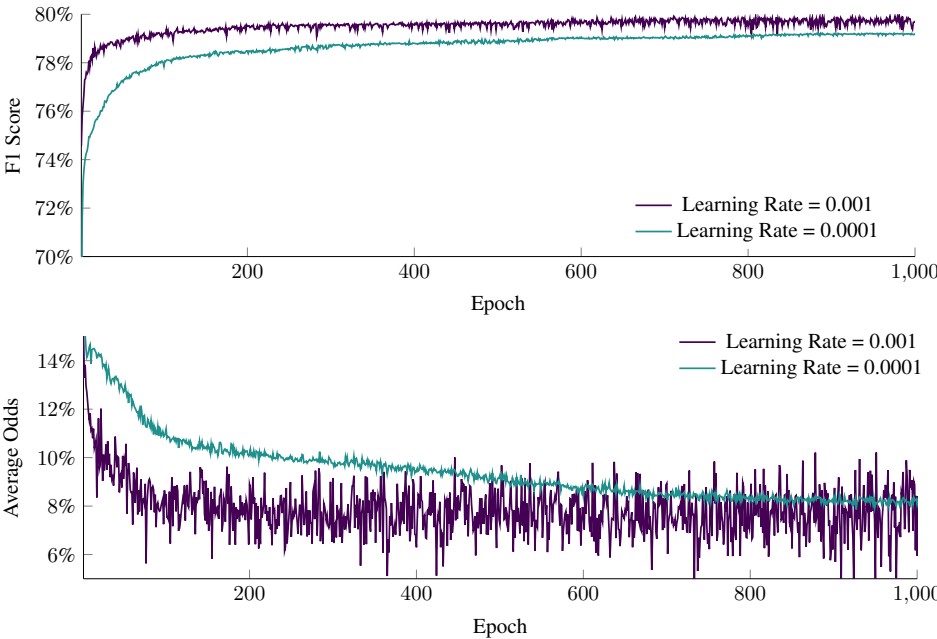

Figure 23: Using a lower learning rate can achieve reasonably stable fairness scores, however the convergence speed is significantly slower. Moreover, the lower learning rate does not convergence to the same accuracy scores, and looses a noticeable margin of accuracy.

### F.2 CHANGING PREDICTIONS AND DATA DISTRIBUTION

We note changing predictions for all 5 settings described above in Figure 24. Note that since we are focusing on changing training hyperparameters for ACSIncome, the data distribution remains the same as in Figure 4(a) (also copied here in Figure 24 for reference).

Same as before, model instability has increased with smaller batch size, higher learning rate, or bigger model architecture, but the trends of vulnerability for various subgroups remains the same. We already know that the models have not converged for higher batch size or smaller learning rate, but we can still see the same trends of higher vulnerability for minority, although they are not as pronounced as in already converged models.

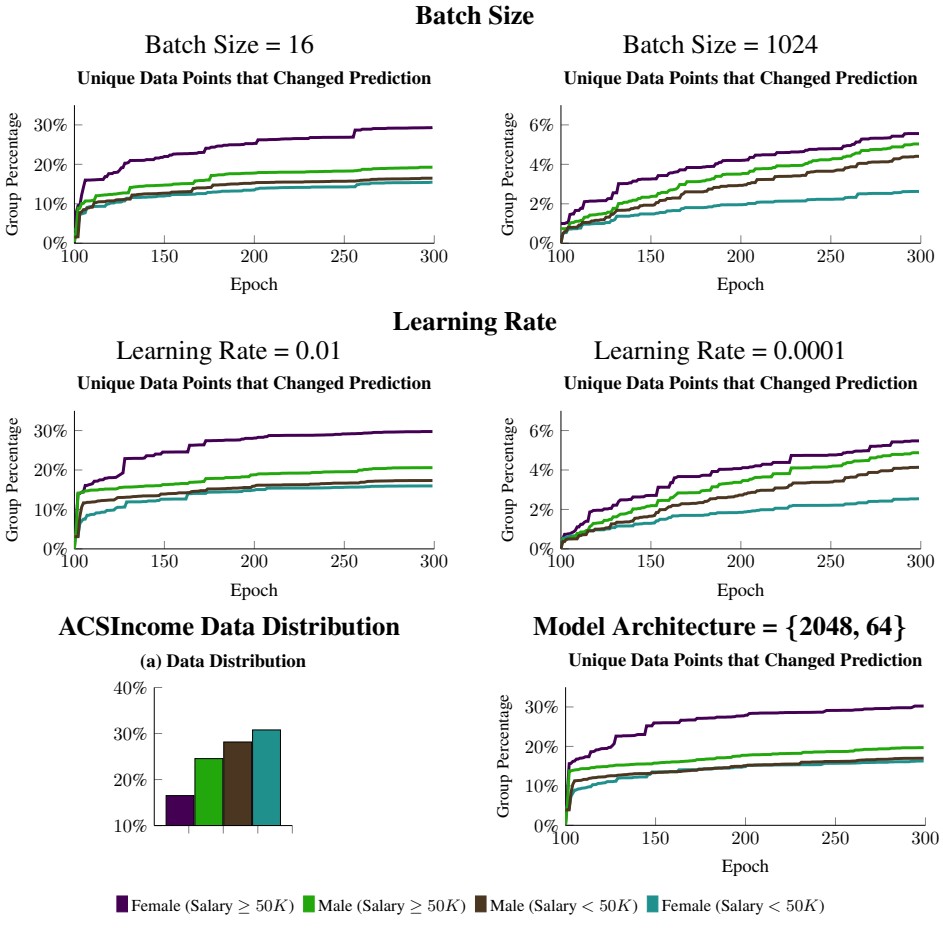

Figure 24: Additional experiments reveal similar trends as seen in Figure 4. These results highlight subgroups with least representation being the most vulnerable to changing predictions.

### F.3 MANIPULATING GROUP LEVEL ACCURACY WITH DATA ORDER

We provide additional results in Figure 25, highlighting the predictability of model fairness based on the data order. Note that the hyperparameter setting for an additional epoch of fine-tuning during group accuracy manipulation is the same as the setup used for training that particular model. For example, when manipulating models which were trained with batch size 16, the single epoch of fine-tuning is also done with batch size 16.

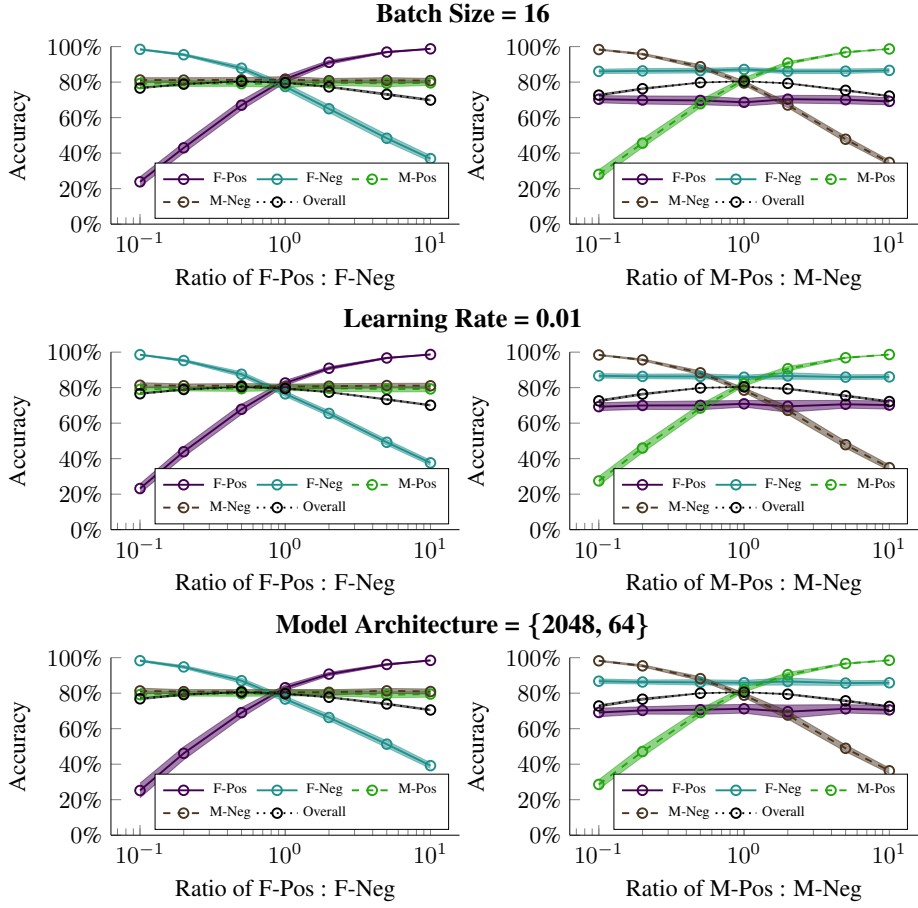

Figure 25: Additional experiments reveal similar trends as seen in Figure 8. In only a single epoch of training, we are able to manipulate the group level accuracy trade-off, with relatively small impact on overall accuracy.

We also provide results for models with high batch size and low learning rate separately in Figure 26. However, note that not only are these models not converged, but they are also fine-tuned on the same inefficient hyperparameter settings. Thus, the trends here are not comparable, but added for completeness.

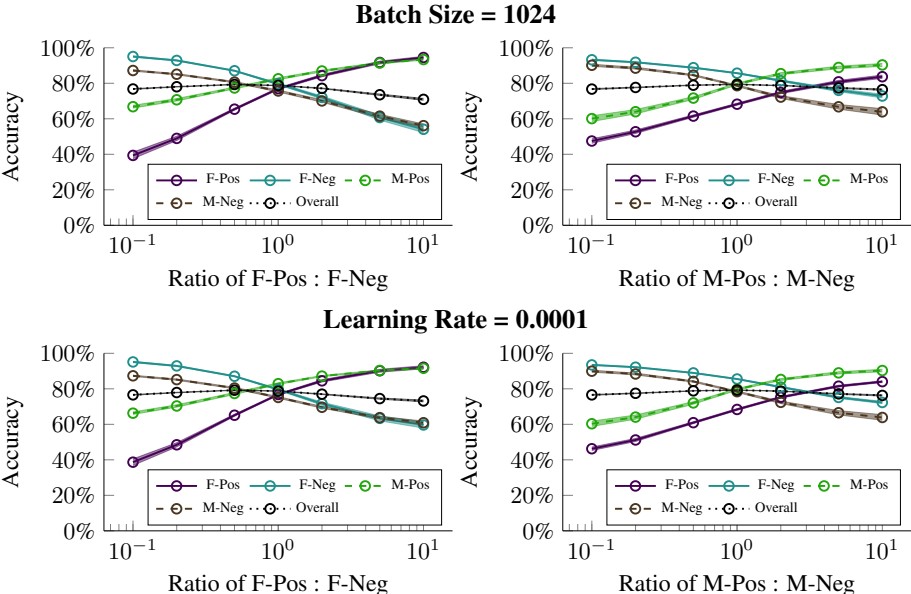

Figure 26: Additional experiments as seen in Figure 25, but for models that did not converge as noted earlier. The results are added for completeness.

# G  ADDITIONAL EXPERIMENTS FOR OTHER FAIRNESS METRICS

We repeat the experiment in the main text for two more fairness metrics, Equal Opportunity (EOpp) and Disparate Impact (DI) Hardt et al. (2016)

For model predictions $R$, true labels $Y$, and sensitive attributes $A$, EOpp can be defined as,

$$EqualOpportunity := |\mathbb{P}(R = 1|Y = 1, A = 0) - \mathbb{P}(R = 1|Y = 1, A = 1)|. \tag{2}$$

Similarly, DI can be defined as,

$$DisparateImpact := 1 - min(\frac{\mathbb{P}(R = 1|A = 0)}{\mathbb{P}(R = 1|A = 1)}, \frac{\mathbb{P}(R = 1|A = 1)}{\mathbb{P}(R = 1|A = 0)}). \tag{3}$$

### G.1 High Variance in Fairness Scores

We start by repeating the results in Figure 1 to show high variance in different fairness measures.

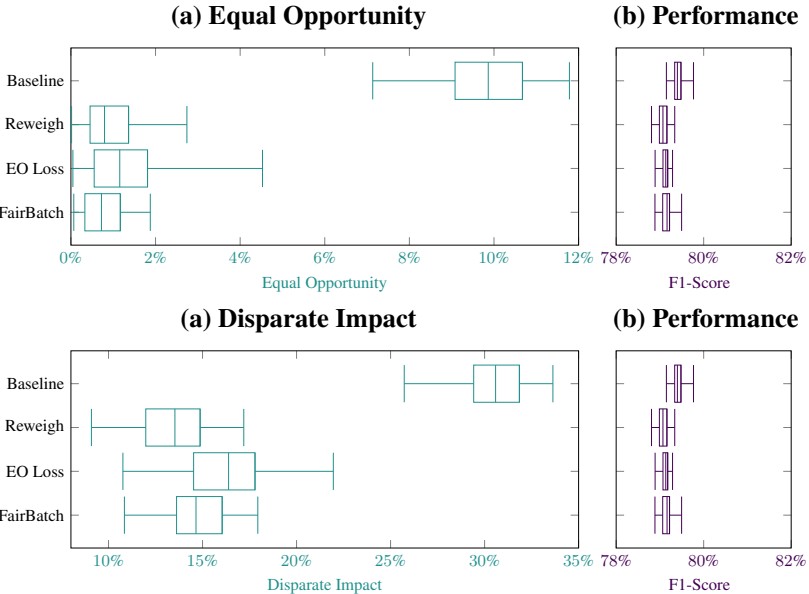

Figure 27: Additional experiments with different fairness metrics for setting in Figure 1. Clearly, fairness has a high variance across identical runs. Note that the x-axis across fairness and accuracy metric is not similarly scaled for disparate impact to keep the results readable.

### G.2 Weight Initialization and Random Reshuffling

We provide additional results for EOpp and DI for the experiment conducted in Figure 3 on ACSIncome dataset. The results collected in Figure 28 show similar trends as seen in the main text.

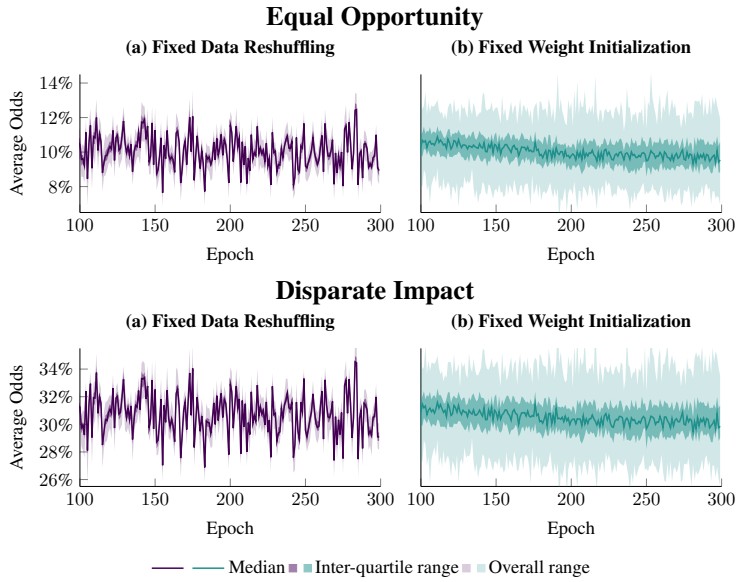

Figure 28: Additional experiments with fairness metric EOpp and DP reveal similar trends as in Figure 3. These results further highlight the dominant impact of random reshuffling on fairness.

## G.3 CAPTURING VARIANCE IN A SINGLE TRAINING RUN

We show the empirical similarity of distribution across multiple training runs and multiple epochs in a single training run for fairness measures EOpp and DI in Figure 29.

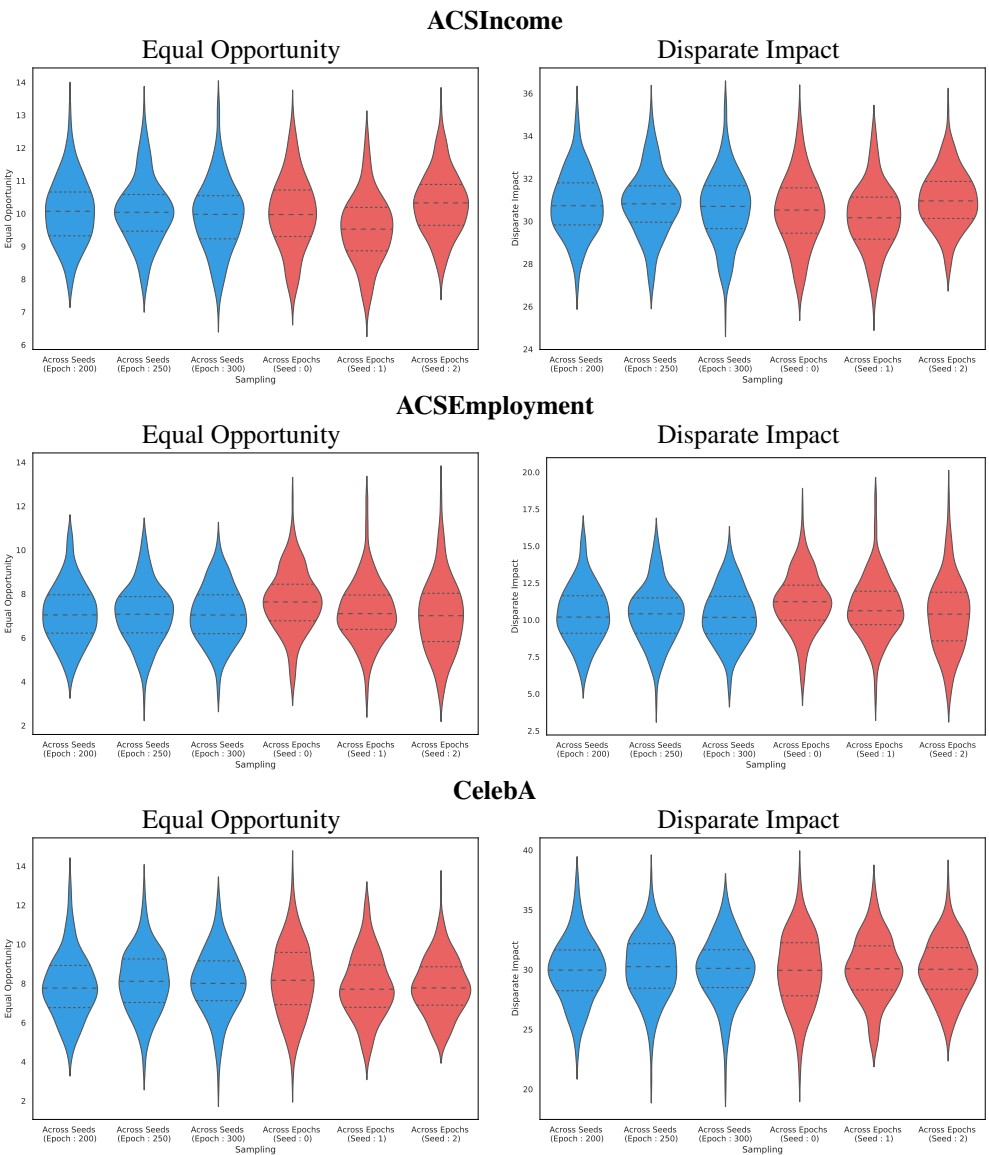

Figure 29: Additional experiments for fairness metrics EOpp and DI. Fairness scores across multiple training runs and across epochs in a single training run have similar empirical distributions. Thus, studying this distribution across epochs provides a highly efficient alternative.

## H    10 RAW TRAINING RUNS FROM FIGURE 3

We plot 10 randomly chosen training runs each for fixed weight initialization and fixed random reshuffling in Figure 30 and Figure 31, respectively. As expected, each individual training run in both settings has high variance across epochs even after convergence. More importantly, the trends of fairness matches closely across multiple runs for fixed random reshuffling, even though they started from different weight initialization.

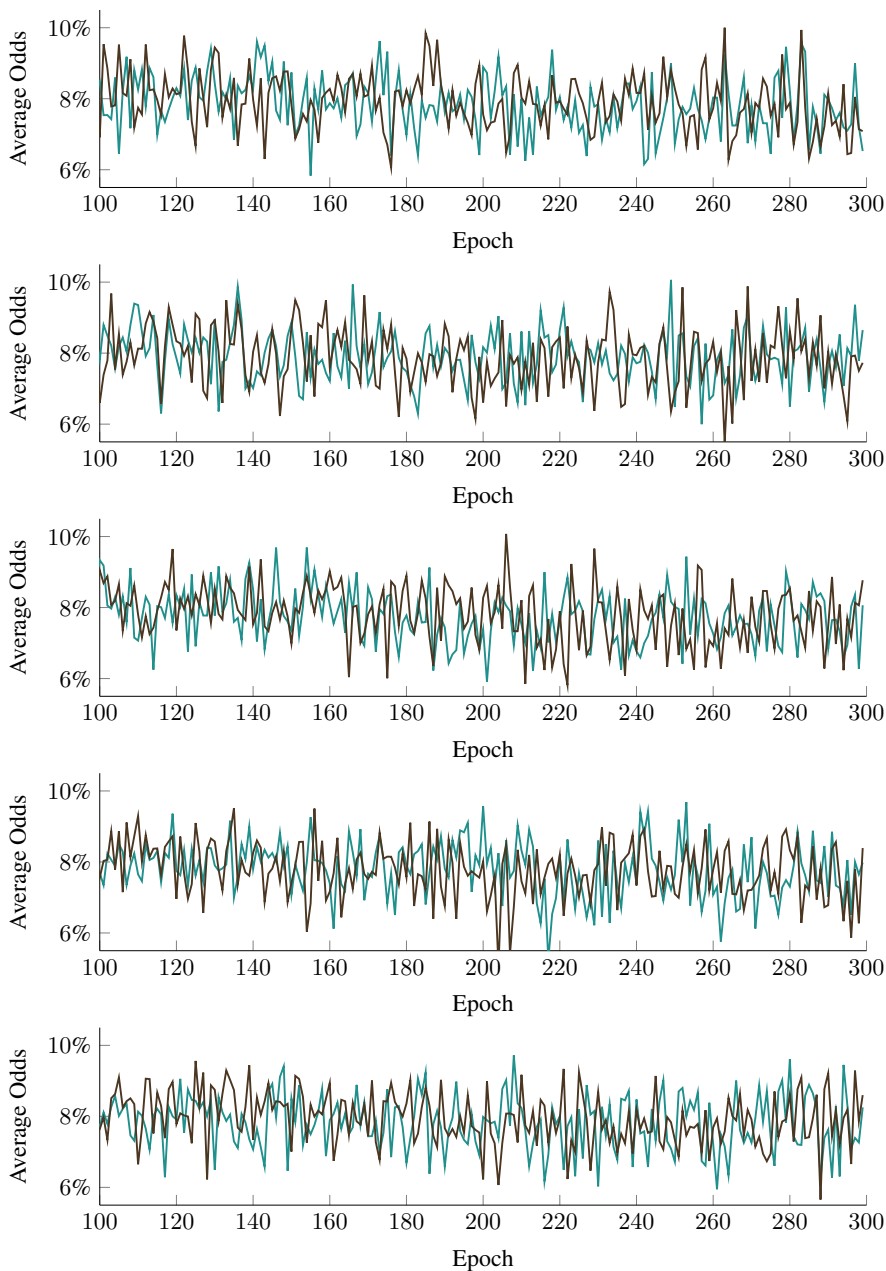

Figure 30: 10 randomly chosen raw training runs plotted in groups of 2 for fixed weight initialization.

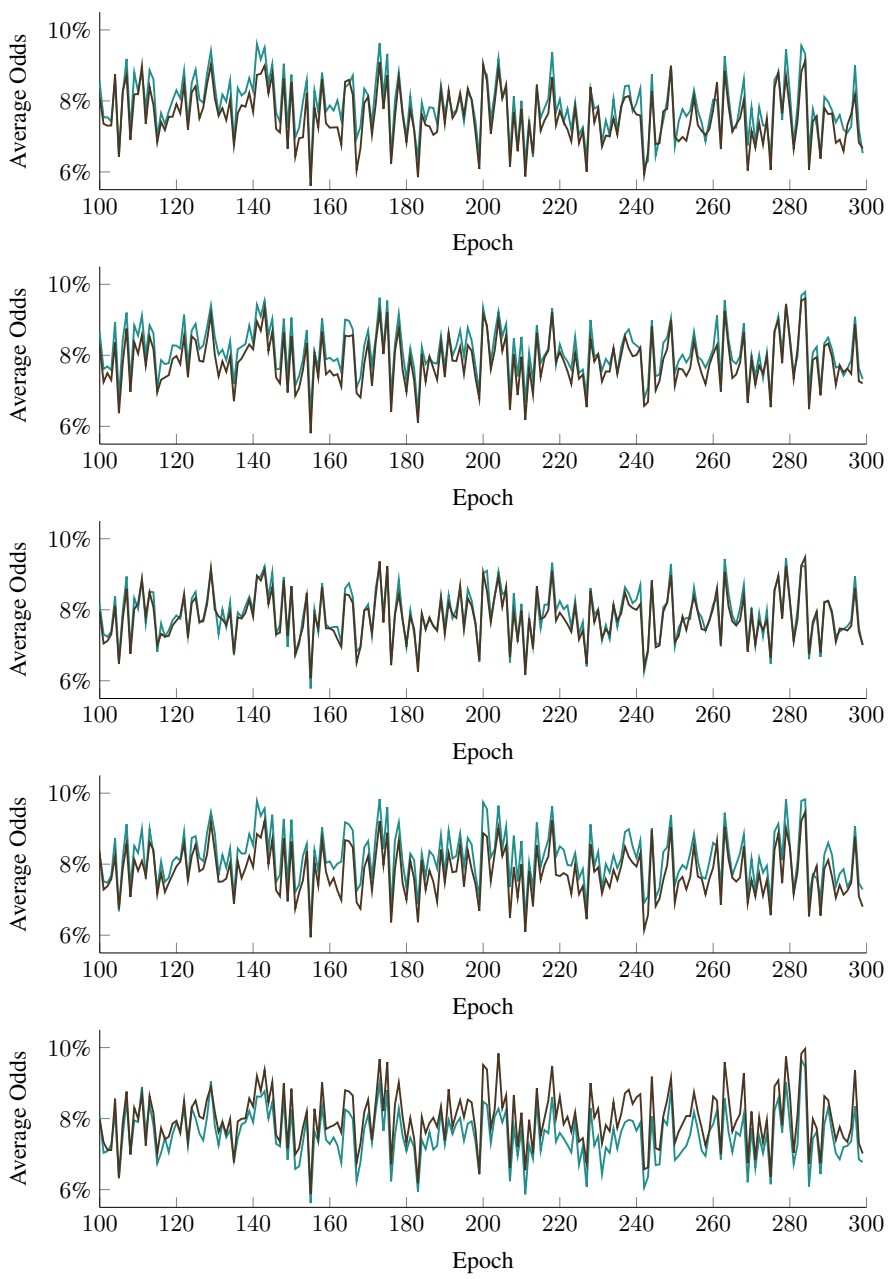

Figure 31: 10 randomly chosen raw training runs plotted in groups of 2 for fixed random reshuffling.

