# OpenReview forum: "On The Impact of Machine Learning Randomness on Group Fairness"
_ICLR.cc/2023/Conference — Submitted to ICLR 2023_

### Official Review · Reviewer_ofwu · 2022-10-23

**Confidence:** 4
**Correctness:** 3
**Technical Novelty And Significance:** 1
**Empirical Novelty And Significance:** 1
**Recommendation:** 3

**Clarity, Quality, Novelty And Reproducibility:**

The paper is clear and the experiments are, in principle, reproducible. I don't think that the work is significantly novel.

**Strength And Weaknesses:**

The authors make an interesting observation. This, however, seems to be only a starting point and more discussion and experiments are needed. I'll detail below some of the reasons behind this consideration.

First, section 3 is titled "Sources of variance" but the only two sources of variance that are considered are weights initialization and random reshuffling. Batch size (which is fixed, I believe), architectures (two, fixed), and possibly other parameters are not discussed. These parameters most likely influence variance. In particular, as batch size increases, random order of the data will play less of a role. Why hasn't batch size been considered?

Second, I would expect data reshuffling to possibly play a role in the initial epochs but less in the later ones when the weights have converged to one of the minima. I'm not sure why this doesn't occur in the experiments. For this, it would be useful to see how overall accuracy increases with the number of epochs. Relatedly, I am very surprised that there is the similar amount of variance for a fixed epoch after 100 and 200 epochs. We could look into that by visualizing the curves for each individual model in figure 3 right, rather than the distribution for a given epoch.

Third, the authors consider three datasets and only one measure fairness measure. Does this behavior generalize to other fairness measures? Why was this fairness measure chosen?

Fourth, section 5 is based on the idea of reweighing the data distribution to change the fairness properties of the model evaluated on the original data distribution. This idea has been widely studied in the literature (both in the fairness and machine learning in general) and I'm not sure that I fully understand where the novelty is, especially how this affects the . The observed behavior may also not even persist as one tunes the batch size and increases the number of epochs (see https://arxiv.org/abs/1812.03372).

A paper that was not cited but is certainly relevant is https://arxiv.org/abs/2107.10171


**Summary Of The Paper:**

The paper investigates how a specific fairness metric varies with the weights initialization and the batches data ordering used in the training of neural networks on three datasets. The authors find that the order in which data observations are fed to the networks during training significantly affects the variance in the fairness metric and conduct experiments to corroborate this claim.

**Summary Of The Review:**

The paper builds upon an interesting observations but more work is needed in order to support the conclusions drawn in the paper (e.g., those at the end of section 4.2).

---

> ### Author Response · Authors · 2022-11-19
> **Response to Reviewer ofwu (Part 2 of 2)**
>
> **...Continued**
>
> > I would expect data reshuffling to possibly play a role in the initial epochs but less in the later ones when the weights have converged to one of the minima. I'm not sure why this doesn't occur in the experiments. For this, it would be useful to see how overall accuracy increases with the number of epochs.
>
> The reviewer's intuition on the model performance stabilizing in later epochs is indeed true for accuracy. However, that is not the case for fairness measures. As discussed in the previous comment, models trained on imbalanced data have disparate stability on various subgroups, which in turn will be reflected in any fairness measure defined on the output of this model. We have added this discussion in the revised version to further clarify the intuition.
>
> We thank the reviewer for pointing out the missing overall training curve, and we have now added the same in the revised version (Appendix C). The results show that the model maintains a stable overall accuracy, despite high variance in fairness scores. Stable accuracy does not imply that the underlying predictions do not change significantly, but that the changes are balanced overall such that they have minimal impact on the accuracy despite a disparate impact on individual groups.
>
> > Relatedly, I am very surprised that there is the similar amount of variance for a fixed epoch after 100 and 200 epochs. We could look into that by visualizing the curves for each individual model in figure 3 right, rather than the distribution for a given epoch.
>
> As can be seen in the training curve now added in the revised version (Appendix C), the model converges before epoch 100, and thus the change in model behavior beyond that is simply movement in the local basin. This is due to the higher instability of predictions on the minority (Figure 4). As requested by the reviewer, we have now added 10 individual training runs each in both settings of Figure 3 separately in the revised version (Appendix H). The trends follow our expectation and show no signs of dampening in fairness variance as the training continues.
>
> > The observed behavior may also not even persist as one tunes the batch size and increases the number of epochs (see https://arxiv.org/abs/1812.03372)
>
> We discussed the impact of changing batch size in the previous comment, and more details can be found in Appendix F. We have now also added the fairness variance across epochs for a model trained up to 3000 epochs (instead of the standard 300 in our paper) in the revised version, to show that the variance does persist despite increasing the number of training epochs (Appendix E). The reasoning here is the same as that for similar variance at epochs 100 and 200 as discussed above.
>
> Finally, the reference provided by the reviewer is quite interesting, but the observations in that work are conditional to settings where the model can perfectly classify the training data, i.e. fits to a 100% training accuracy. That, however, is not the case for any of our settings.
>
> > A paper that was not cited but is certainly relevant is https://arxiv.org/abs/2107.10171
>
> We thank the reviewer for pointing out the missing reference. Leave one out fairness is an interesting perspective of fairness from the lens of differential privacy and we have added a brief discussion for the same in our revised version.
>
> **References :**
> - Esther Rolf, Theodora T. Worledge, Benjamin Recht, and Michael Jordan. "Representation matters: Assessing the importance of subgroup allocations in training data." In International Conference on Machine Learning, pp. 9040-9051. PMLR, 2021.

---

> ### Author Response · Authors · 2022-11-19
> **Response to Reviewer ofwu (Part 1 of 2)**
>
> We thank the reviewer for their feedback. We have improved our paper based on their comments. The changes in the revised version are marked in red. We also provide clarification on individual comments below.
>
> > section 3 is titled "Sources of variance" but the only two sources of variance that are considered are weights initialization and random reshuffling. Batch size (which is fixed, I believe), architectures (two, fixed), and possibly other parameters are not discussed. These parameters most likely influence variance. In particular, as batch size increases, random order of the data will play less of a role. Why hasn't batch size been considered?
>
> Various factors in neural model training that impact fairness variance can either be stochastic or non-stochastic. In our work, we focus specifically on the impact of stochastic choices in the training algorithm, i.e. weight initialization, random reshuffling, and dropout regularization (added in the revised version Appendix D). We have also updated the title of Section 3 to "Sources of Randomness" to avoid further confusion.
>
> However, the reviewer is correct to point out that even the non-stochastic choices can indeed impact the fairness variance observed. We have now added new results in the revised version under changing batch size, learning rate, and architecture (Appendix F). More specifically, increasing the batch will indeed dampen the impact of noise in mini-batch gradient descent (we can also achieve the same with a lower learning rate). But this noise is vital to the model's convergence, and the larger batch size not only slows down the convergence significantly but even converges the model to a worse accuracy score. Thus, the ideal batch size is the one that can help the model reach a better minima in less number of epochs, but such a setting will be accompanied by high fairness variance that we study in our work. More details on this discussion can be found in the revised version (Appendix F).
>
> > Does this behavior generalize to other fairness measures? Why was this fairness measure chosen?
>
> Models trained with a heterogenous combination of data from multiple groups can be expected to have disparate uncertainty in their predictions for these groups. We show that neural model predictions are more unstable on minority subgroups, i.e., subgroups with smaller representation in the overall dataset (Figure 4). As the underlying model is more unstable for a particular subgroup, any fairness measure defined on the output of this model will also reflect this instability (Figure 1; Figure 27). Finally, we also show that our data order manipulation can control group-level accuracies which would be the basis of any subsequent fairness measure (Figure 8).
>
> We have added more results in the revised version for fairness measures Equal Opportunity (EOpp) and Disparate Impact (DI) (Appendix G) to support our claim with empirical results.
>
> > section 5 is based on the idea of reweighing the data distribution to change the fairness properties of the model evaluated on the original data distribution. This idea has been widely studied in the literature (both in the fairness and machine learning in general) and I'm not sure that I fully understand where the novelty is, especially how this affects the .
>
> The idea of manipulating data order in Section 5 is not synonymous with reweighing data distribution, as the model is still trained on the whole dataset in every epoch, i.e., the distribution of the complete epoch is still the same and has not been 'reweighed'. It's only the distribution of the suffix of the data order, or in other words, the most recent gradient updates seen by the model, that has the desired distribution. It's the immediate impact of only the last few batches which is surprising and a novel finding of our work. Moreover, explicit reweighing of data to simulate a change in data distribution is connected with variance (Rolf et al., 2021), which is not an expected trend of data reordering. Finally, existing literature on reweighing in fair machine learning relies on training the model from scratch with a new reweighed data distribution, while we show that there is a fair model 'near' an already converged model which was trained without any fairness constraints, i.e., we motivate the possibility of fair fine-tuning.
>
> The last part of the sentence in the review got cut off, so we are not sure what the reviewer was alluding to.
>
> **Continued...**

---

### Official Review · Reviewer_XbQB · 2022-10-24

**Confidence:** 2
**Correctness:** 3
**Technical Novelty And Significance:** 2
**Empirical Novelty And Significance:** 2
**Recommendation:** 5

**Clarity, Quality, Novelty And Reproducibility:**

The paper is easy to follow, and it is generally well-written. I am not an expert in the area so I cannot assess the novelty of this work. It seems like similar observations have been made in a setting with no fairness consideration. In that sense, observing variance in performance when restricting to a subset of data is not surprising. The authors claim that the code is attached but I do not see a link.

**Strength And Weaknesses:**

---------------------------------
Strengths:
---------------------------------
-- The paper is fairly well-written and it is easy to understand the paper.

---------------------------------
Weaknesses:
---------------------------------
-- The technical contribution is limited. The paper does not provide any algorithms or analysis.

-- While having theory is not always required, I find the empirical analysis of the paper to be insufficient. The paper mainly focuses on one dataset and there should be more empirical evidence to justify the provided solution. More intuition about the proposed solution would also be appreciated.

-- I may have not fully misunderstood the paper, but what is the rationale behind lowering the variance besides the training time? Can the results be repeated and averaged to lower the variance?

---------------------------------
Disclaimer:
---------------------------------
-- I do not an expert on the topic and it is possible that I have missed the major points of the paper.

**Summary Of The Paper:**

The paper observes that the performance of a deep learning model might exhibit a high variance between different training instances and hence this might lead to a high variance in the performance gap across different groups. The paper proposes a solution based on the reordering of data, to lower the variance.

**Summary Of The Review:**

In summary, this paper is outside my area of expertise but based on what I understand, the contributions and novelty are limited.

----------------------------------------
Post Rebuttal:
----------------------------------------
I want to thank the authors for their detailed responses. I have read the other reviews and the rebuttal and I stand by my original assessment.

---

> ### Author Response · Authors · 2022-11-19
> **Response to Reviewer XbQB (Part 2 of 2)**
>
> **...Continued**
>
> > It seems like similar observations have been made in a setting with no fairness consideration. In that sense, observing variance in performance when restricting to a subset of data is not surprising.
>
> High variance in fairness scores is not our contribution, and has been previously observed and highlighted in fairness literature (Qian et al., 2021; Amir et al., 2021; Sellam et al., 2021; Soares et al., 2022). However, these works focus on the impact of high variance and the lack of reliability in results that do not account for variance due to randomness in model training. We instead focus on trying to understand the origin of fairness variance and trace it to the underlying instability in minority predictions. Moreover, we investigate the intimate and immediate relationship of model fairness with the training data order and the possibility of manipulating the data order to indirectly control group accuracy. To the best of our knowledge, we are not aware of any such work in literature.
>
> > The paper mainly focuses on one dataset and there should be more empirical evidence to justify the provided solution.
>
> We provide our experiments on three different datasets as noted in Section 2. While we focus on one primary dataset in the main text (i.e. ACSIncome), all our experiments are repeated for all three datasets in the appendix (Appendix E). Moreover, based on reviewer suggestions, we have now added more results in the appendix under changing training settings, i.e., batch size, learning rate, and architecture (Appendix F), as well as across various fairness measures (Appendix G), to further diversify the experiment landscape.
>
> > The authors claim that the code is attached but I do not see a link.
>
> The code is attached as a supplementary zip file and not linked directly to any external site to preserve anonymity.
>
> **References :**
> - Ioana Baldini Soares, Dennis Wei, Karthikeyan Natesan Ramamurthy, Moninder Singh, and Mikhail Yurochkin. Your fairness may vary: Pretrained language model fairness in toxic text classification. In Annual Meeting of the Association for Computational Linguistics, 2022.
> - Thibault Sellam, Steve Yadlowsky, Ian Tenney, Jason Wei, Naomi Saphra, Alexander D’Amour, Tal Linzen, Jasmijn Bastings, Iulia Raluca Turc, Jacob Eisenstein, et al. The multiberts: Bert reproductions for robustness analysis. In International Conference on Learning Representations, 2021.
> - Shangshu Qian, Viet Hung Pham, Thibaud Lutellier, Zeou Hu, Jungwon Kim, Lin Tan, Yaoliang Yu, Jiahao Chen, and Sameena Shah. Are my deep learning systems fair? an empirical study of fixed-seed training. Advances in Neural Information Processing Systems, 34, 2021.
> - Silvio Amir, Jan-Willem van de Meent, and Byron C Wallace. On the impact of random seeds on the fairness of clinical classifiers. In Proceedings of the 2021 Conference of the North American Chapter of the Association for Computational Linguistics: Human Language Technologies, pp. 3808–3823, 2021.
> - Felix Draxler, Kambis Veschgini, Manfred Salmhofer, and Fred Hamprecht. "Essentially no barriers in neural network energy landscape." In International conference on machine learning, pp. 1309-1318. PMLR, 2018.
> - Timur Garipov, Pavel Izmailov, Dmitrii Podoprikhin, Dmitry P. Vetrov, and Andrew G. Wilson. "Loss surfaces, mode connectivity, and fast ensembling of dnns." Advances in neural information processing systems 31 (2018).
> - Samuel K. Ainsworth, Jonathan Hayase, and Siddhartha Srinivasa. "Git re-basin: Merging models modulo permutation symmetries." arXiv preprint arXiv:2209.04836 (2022).

---

> ### Author Response · Authors · 2022-11-19
> **Response to Reviewer XbQB (Part 1 of 2)**
>
> We thank the reviewer for their feedback. We have improved our paper based on their comments. The changes in the revised version are marked in red. We also provide clarification on individual comments below.
>
> > More intuition about the proposed solution would also be appreciated.
>
> - Prediction Instability in Minorities: Models trained with a heterogenous combination of data from multiple groups can be expected to have disparate uncertainty in their predictions for these groups. We show that neural model predictions are more unstable on minority subgroups, i.e., subgroups with smaller representation in the overall dataset (Figure 4). As the underlying model is more unstable for a particular subgroup, any fairness measure defined on the output of this model will also reflect this instability (Figure 1; Figure 27). Thus, fairness variance in existing literature (Amir et al., 2021; Sellam et al., 2021; Soares et al., 2022) is mainly due to the prediction instability of the learning algorithm on under-represented groups.
> - Immediate Impact of Data Order: We trace the source of instability in under-represented groups to the data order fed to the model during training, and show an immediate impact of changing the data order on model fairness (Figure 5). Two existing lines of research can explain this behavior, (i) the presence of no energy valleys between minimas of separately trained models (Draxler et al., 2018; Garipov et al., 2018), which could allow converged models to easily move between separate minimas, and (ii) the existence of permutation symmetries between various minima basins, such that each converged model has a functionally equivalent copy in every other basin (Ainsworth et al. 2022), and thus models fed the same data order might move towards the same functionally equivalent model, if not the exact same weight vector.
> - Manipulating Group Accuracy with Data Order: Finally, we use the immediate impact of data order to manipulate group-level accuracy. We hypothesize that by changing the data distribution of the most recent batches seen by the model, we can temporarily reshape the loss landscape and allow the model to move toward the desired distribution. We found this algorithm effective (Figure 8), and we even proposed special case distributions that can rival existing bias mitigation methods (Figure 9).
>
> We have rewritten parts of our paper to add appropriate intuitions, and hopefully remove any confusion.
>
> > what is the rationale behind lowering the variance besides the training time? Can the results be repeated and averaged to lower the variance?
>
> Yes, reducing the training time is indeed the central motivation for our discussion in Section 4.2. We found a comparable quality of black swans captured by sampling across multiple training runs and sampling inside a single training run in Figure 7(b), even though the latter will require 50 times less computation. We also observed similar results for the overall fairness score distribution (Figure 7(a)).
>
> Running multiple identical training instances is a big problem for the community. For example, a recent work (Sellam et al., 2021) released 25 separate training runs of pre-trained language model BERT, while only changing the random seeds, which took them a few months of continuous training on 16 Cloud TPUs. This can be a dangerous precedent to set for future work on model fairness. While the need to incorporate the impact of randomness in training has been repeatedly shown in recent literature (Amir et al., 2021; Sellam et al., 2021; Soares et al., 2022), we emphasize that running multiple identical training instances is not an efficient solution for the same.
>
> **Continued...**

---

### Official Review · Reviewer_Qe5p · 2022-11-05

**Confidence:** 2
**Correctness:** 3
**Technical Novelty And Significance:** 2
**Empirical Novelty And Significance:** 3
**Recommendation:** 5

**Clarity, Quality, Novelty And Reproducibility:**

Clarity
- sec 2 first para. Could please explain these datasets in the appendix so that readers do not have to hunt in numerous different works? The datasets are important to the empirical evaluations, and the additional description will make the work more modular.
- description of the experimental setup is clear
- sec 3.2 Can you be exact in what it means for predictions to change? Does this mean that before including the data instance, the model would have predicted the instance incorrectly? In light of this confusion I am unable to understand Figure 4b.
- sec 4 is well-written and has a nice experimental setup

Novelty: I am unfamiliar with work related to this paper but did a quick search on arXiv, and did not find similar work.

Reproducibility: I believe I could reproduce this work if need be---enough experimental detail has been given throughout the work and in the appendices.

Small suggestions:
- section 6 second-to-last line: "practise of" to "practice of"
- section 5 first para last line: "and place at the" to "and placed at the"
- missing period eqn (1)


**Strength And Weaknesses:**

Strengths
- The topic of this work is important, i.e., what are some of the causes of high variance in fair deep learning?
- The concept of rearranging data instances to change sub-group performance is technically interesting.

Weaknesses
- There are other sources of randomness besides weight initialization and the training data, e.g., various regularization techniques and train/fit/validation sets (as the authors mentioned in the text). Can the authors provide reasoning for why only weight init and data ordering are considered in this work?
- Unfortunately, I do not understand section 3.2 (see `clarity' section for questions)

**Summary Of The Paper:**

This paper tries to find empirically validated reasoning for the cause of high variance in fair deep learning, where the fairness metric of interest is average odds (AO), i.e., the average disparity between true and false positive rates. This is generally attributed to randomness (what they call non-determinism) in training, e.g., randomness in data order and weight initialization. A series of empirical evaluations are conducted that investigate the source of this variance. The following assertions are made in light of the empirical evidence:
-  Variance (wrt to AO) can be attributed more to data shuffling than to weight initialization.
- A model's fairness is predictable based on the most recent training points.
The work also asserts that a proxy for understanding model fairness across runs can be variance in fairness across epochs. Moreover, the work shows that sub-group performance on the fairness metric can be manipulated by rearranging the training data instances.


**Summary Of The Review:**

The paper empirically shows that the order of data instances in model training can impact the quality of the averaged odds fairness metric, introduces variance in fairness performance across epochs as a proxy for understanding fairness across runs, and shows that sub-group performance on the fairness metric can be manipulated by rearranging the training data instances. Some aspects of the experimental evaluation related to model predictability and recent data instances are unclear.

---

> ### Author Response · Authors · 2022-11-19
> **Response to Reviewer Qe5p**
>
> We thank the reviewer for their feedback. We have improved our paper based on their comments. The changes in the revised version are marked in red. We also provide clarification on individual comments below.
>
> > There are other sources of randomness besides weight initialization and the training data, e.g., various regularization techniques and train/fit/validation sets (as the authors mentioned in the text). Can the authors provide reasoning for why only weight init and data ordering are considered in this work?
>
> Randomness due to the training algorithm and randomness due to input data are considered two distinct sources of randomness in neural model training, and have been studied separately in the literature (Zhuang et al., 2022). In our work, we focus on randomness due to the training algorithm and thus do not consider the randomness introduced by the train/val/test split.
>
> We thank the reviewer for pointing out another source of randomness in the training algorithm that we missed, i.e. regularization. We extend our discussion and introduce dropout regularization in our training pipeline. Randomness in dropout regularization even under fixed weight initialization and fixed random reshuffling introduces minute variance across multiple runs. While it adds noise to previously noted trends, the variance in fairness across consecutive epochs in a single training run, as well as the dominance of random reshuffling on fairness scores is still clearly noticeable (Figure 12). More details about this discussion are now added in the revised version (Appendix D).
>
> > Can you be exact in what it means for predictions to change? Does this mean that before including the data instance, the model would have predicted the instance incorrectly? (Ref : Section 3.2)
>
> As we are concerned with the fairness of the final decisions made by the model, we focus on a change in the model's final discrete output class when discussing changing predictions. More specifically, a model is said to have undergone a change in prediction for some input $x$ during epoch $i$, if $\hat{y}\_i(x) \neq \hat{y}\_{i-1}(x)$, where $\hat{y}\_i(x)$ is the output class when passing the input $x$ through the model checkpoint at the end of epoch $i$. In Section 3.2, we isolate all data points which change their prediction at least once between epochs 100 and k. This can also be interpreted as picking out all data points which didn't maintain the same output class at the end of every epoch between epochs 100 and k, or in other words, flipped their prediction class at least once (only for binary classification).
>
> To further clarify, the training dataset does not change during training, i.e. the model is trained on the complete training dataset at every epoch. The language 'passing a data point through the model' refers to inference on that particular data point. The change in model behavior is simply the change due to the random reshuffling of the data order during training. We thank the reviewer for pointing out the lack of clarity and we have updated Section 3.2 in the revised version to hopefully remove any confusion.
>
> > Could please explain these datasets in the appendix so that readers do not have to hunt in numerous different works?
>
> We thank the reviewer for the suggestion. We have now added details about all three datasets that we use in our work in the revised version (Appendix B).
>
> > Small suggestions
>
> We have corrected all syntactical errors as pointed out by the reviewer.
>
>
> **References :**
> - Donglin Zhuang, Xingyao Zhang, Shuaiwen Song, and Sara Hooker. "Randomness in neural network training: Characterizing the impact of tooling." Proceedings of Machine Learning and Systems 4 (2022): 316-336.

---

### Decision · Program_Chairs · 2023-01-20

**Decision:**

Reject

**Justification For Why Not Higher Score:**

As discussed by reviewers, ultimately I found the experimental setup somewhat ad-hoc, more of an exploratory exposition. It is difficult to interpret variability to one metric without a sense of how much is inherent to the data and how much is inherent to the algorithm. At least I would expect to anchor results on some other learning method with fewer degrees of freedom like XGBoost and try to make sense how much do we even still gain by worrying about deep learning issues of model fitting (for the Folktables examples in particular, I would not be shocked if a generalized linear model is essentially as good as it gets). Also, much of the phenomena observed may boil down to group imbalance in the data, which is not exactly surprising and mitigated by stratified sampling in the mini-batches. I do think this type of study has value, but at the present stage it feels some points open that should be treated earlier than later.

**Justification For Why Not Lower Score:**

N/A

**Metareview: Summary, Strengths And Weaknesses:**

Uncertainty and unfairness have a close connection, as underprivileged groups suffer, among other issues, from sources of variability that larger groups may not. The paper breaks down the analysis of some of these sources with a focus on the variability of the estimator of fairness metrics.

Strengths: a presentation of a systematic characterization of sources of variability in the pipeline of a machine learning solution that should produce estimates of some fairness metric, in particular those addressing randomness in the optimization process.

Weaknesses: as discussed by reviewers, ultimately I found the experimental setup somewhat ad-hoc, more of an exploratory exposition. It is difficult to interpret variability to one metric without a sense of how much is inherent to the data and how much is inherent to the algorithm. At least I would expect to anchor results on some other learning method with fewer degrees of freedom like XGBoost and try to make sense how much do we even still gain by worrying about deep learning issues of model fitting (for the Folktables examples in particular, I would not be shocked if a generalized linear model is essentially as good as it gets). Also, much of the phenomena observed may boil down to group imbalance in the data, which is not exactly surprising and mitigated by stratified sampling in the mini-batches. I do think this type of study has value, but at the present stage it feels some points open that should be treated earlier than later.